

# Sea ice break-up and freeze-up indicators for users

# of the Arctic coastal environment

John E. Walsh[1], Hajo Eicken[1], Kyle Redilla[1], Mark Johnson[2]

[1]International Arctic Research Center, University of Alaska Fairbanks, Fairbanks AK 99775 USA

[2]College of Fisheries and Ocean Sciences, University of Alaska Fairbanks, Fairbanks AK 99775 USA

*Correspondence to:* John E. Walsh (jewalsh@alaska.edu)

September 2022

*The Cryosphere*, Revision 2

**Abstract**

The timing of sea ice retreat and advance in Arctic coastal waters varies substantially from year
to year. Various activities, ranging from marine transport to the use of sea ice as a platform for
industrial activity or winter travel, are affected by variations in the timing of break-up and
freeze-up, resulting in a need for indicators to document the regional and temporal variations in
coastal areas. The primary objective of this study is to use locally-based metrics to construct
indicators of break-up and freeze-up in the Arctic/Subarctic coastal environment. The indicators
developed here are based on daily sea ice concentrations derived from satellite passive
microwave measurements. The "day of year" indicators are designed to optimize value for
users while building on past studies characterizing break-up and freeze-up dates in the open
pack ice. Relative to indicators for broader adjacent seas, the coastal indicators generally show
later break-up at sites known to have landfast ice. The coastal indicators also show earlier
freeze-up at some sites in comparison with freeze-up for broader offshore regions, likely tied to
earlier freezing of shallow water regions and areas affected by freshwater input from nearby
streams and rivers. A factor analysis performed to synthesize the local indicator variations
shows that the local break-up and freeze-up indicators have greater spatial variability than
corresponding metrics based on regional ice coverage. However, the trends towards earlier
break-up and later freeze-up are unmistakable over the post-1979 period in the synthesized
metrics of coastal break-up/freeze-up and the corresponding regional ice coverage. The findings
imply that locally defined indicators can serve as key links between pan-Arctic or global
indicators such as sea-ice extent or volume and local uses of sea ice, with the potential to inform
community-scale adaptation and response.
*Key words*: sea ice, Arctic, break-up, freeze-up, ice concentration

## 1. Introduction

Coastal sea ice impacts residents and other users of the nearshore marine environment in various ways. Perhaps most obvious is the fact that non-ice strengthened vessels require ice-free waters for marine transport, which can serve purposes such as resupply of coastal communities, the transport of extracted resources (oil, liquefied natural gas, mined metals), migration of marine mammals (e.g., bowhead whales) and wintertime travel over the ice by coastal residents.  Key metrics for such uses of the nearshore marine environment are the timing of break-up (or ice retreat) in the spring and the timing of freeze-up (or ice advance) in the autumn or early winter.

Sea ice concentration thresholds have been used in various studies to determine the dates of sea ice opening, retreat, advance and closing (Markus et al., 2009; Johnson and Eicken, 2016; Bliss and Anderson; 2018; Peng et al., 2018; Bliss et al., 2019; Smith and Jahn, 2019). An emerging tendency in these and similar studies is the definition of break-up date as the date on which ice concentration drops below a prescribed threshold and remains below that threshold for a prescribed minimum duration (chosen to eliminate repeated crossings of the concentration threshold as a result of temperature- or wind-driven changes in ice coverage in response to transient weather events). A corresponding criterion is used for the freeze-up date.

Coastal regions present special challenges in the application of such criteria.  First, landfast or shorefast ice (stationary sea ice held in place along the shoreline as a result of grounding and/or confinement by the coast) is common in waters immediately offshore of the coast, particularly in areas with shallow water.  Landfast ice provides especially important sea ice services because it offers a stable platform for nearshore travel, serves as a critical habitat for marine mammals such as seals and polar bears (Dammann et al., 2018), and provides a buffer

against coastal storms (Hosekova et al., 2021).  Landfast ice extends offshore by hundreds of
meters to many tens of kilometers. Figure 1 shows the geographical distribution of landfast ice
in terms of the maximum extent during June for the period 1972-2007.  Landfast ice is most
extensive over shallow waters of the Siberian Seas and the Canadian Archipelago.  Given its
widespread presence at coastal sites in the Arctic, landfast ice will be a key feature in our
assessment of any differences in the sea-ice indicators, particularly for ice break-up, when
comparing coastal to offshore regions.

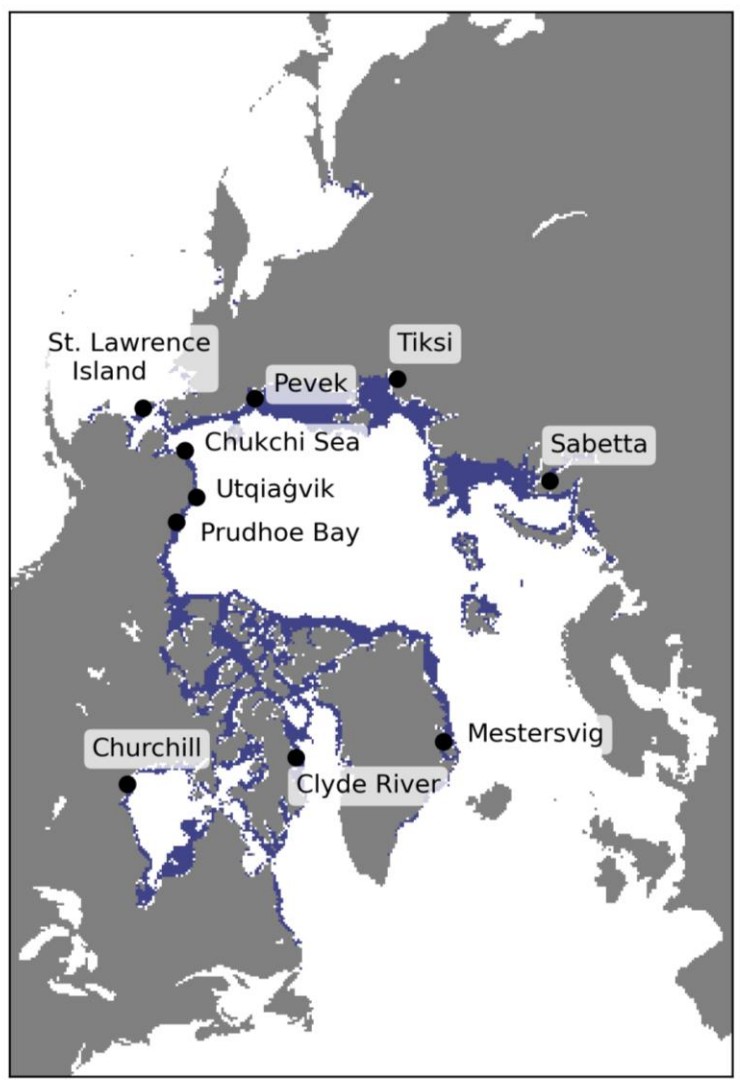


Figure 1.  Landfast ice distribution shown as the maximum extent of landfast ice over the
1972-2007 period.  Data source: National Ice Center via National Snow and Ice Data Center,
NSIDC dataset G02172 -- https://nsidc.org/data/G02172 (accessed 4 September 2022).
A second challenge associated with coastal regions is that sea ice concentrations derived from
passive microwave measurements are prone to contamination by microwave emissions from
land in coastal grid cells. Additionally, many parts of the Arctic coastline have inlets, river
deltas and barrier islands that are not captured by the 25 km resolution of the passive
microwave product. While higher-resolution datasets permitting finer resolution of coastal sea
ice are available from sensors such as AMSR (Advanced Microwave Scanning Radiometer),
the record lengths are sufficiently shorter (about 20 years for AMSR) that trend analyses are
limited by a reliance on such products.  Trend analysis is one of the main components of the
present study.
A pervasive finding from recent studies of trends in Arctic sea ice is a shortening of the sea
ice season.  This finding is often presented in terms of the corresponding lengthening of the
open water season (e.g., Stroeve et al., 2014; Stroeve and Notz, 2018; Onarheim et al., 2018;
Bliss and Anderson, 2018; Peng et al., 2019; Smith and Jahn, 2019).  Because the reduction of
ice extent has been greater in summer than in winter, the percentage of the Arctic sea ice
cover experiencing break-up and freeze-up (i.e., the percentage of the maximum ice cover that
is seasonal) has increased from about 50% in 1980 to more than 70% in recent years
(Druckenmiller et al., 2021; Thomson et al., 2022). Since 1980, the length of the open water
period has increased by between one and two months (over 10 days per decade)
(Stammerjohn et al., 2012; Peng et al., 2019; Thomson et al., 2022), with contributions of
comparable magnitude from earlier break-up and later freeze-up. Regional variations of these
trends, both in the vicinity of the coasts and in regions farther offshore, are the focus of this
paper as well as Bliss et al. (2019), to which we will compare our results.
Trends in freeze-up have been shown previously to be sensitive to the criterion for freeze-up
(Peng et al., 2018; Bliss et al., 2019).  For example, Peng et al. (2018) found that the trends in
the autumn crossing of the 80% concentration were greater than trends in the crossing of the
15% threshold (Thomson et al., 2022), implying a slowing of the autumn/winter ice advance.
Such findings, as well as those of Johnson and Eicken (2016), motivate our use of separate
indicators for the start and end of break-up and freeze-up.
The delayed autumn freeze-up is a manifestation of the release of increased amounts of heat
stored in the upper layers of the ocean, largely as a result of the increased solar absorption
made possible by the earlier break-up.  In this respect, trends in break-up and freeze-up are
intertwined.  This linkage has been demonstrated quantitatively by Serreze et al. (2016) and
Stroeve et al. (2016), who explored the use of break-up timing as a predictor of the timing of
ice advance in the Chukchi Sea and the broader Arctic, respectively.
The primary objective of this study is to use the locally-based metrics to construct indicators
of break-up and freeze-up on Arctic/Subarctic coastal environments.  A secondary objective is
to contribute to efforts at the national and global scale to establish key sets of indicators that
support sustained assessment of climate change and inform planning and decision-making for
adaptation action (AMAP, 2018; IPCC, 2022). At the global, pan-Arctic, and U.S. national
levels, indicators associated with the state of the sea ice cover so far have focused on the
summer minimum and winter maximum extent and ice thickness (IPCC, 2022; AMAP, 2017;
Box et al., 2019; USGCRP, 2017). As outlined by Box et al. (2019), this approach has been
motivated by the objective of describing and tracking the state of key components of the
global climate system. However, large-scale (pan-Arctic) measures of e.g., sea-ice extent or
volume are of little value and relevance to those needing to adapt or respond to such change at
the community or regional scale. Here, we examine the timing of sea-ice freeze-up and break-
up as key constraints for a range of human activities and ecosystem functions in Arctic
settings.
**2.   Data and methods**
The primary data source is the archive of gridded daily sea ice concentrations derived from
the SMMR, SSM/I and SSMIS sensors onboard the Nimbus-7 and various DMSP satellites
dating back to November, 1978. The dataset is NSIDC-0051 of the National Snow and Ice
Data Center (NSIDC) and is accessible at https://nsidc.org/data/nsidc-0051. In the
construction of this dataset, the NASA Team algorithm (Cavalieri et al., 1984) was used to
process the microwave brightness temperatures into a consistent time series of daily sea ice
concentrations. The data are on a polar stereographic grid projection with a grid cell size of 25
km x 25 km. Prior to computing the break-up and freeze-up metrics described below, the data
were processed with a linear interpolation to fill in missing daily values, followed by a spatial
and then temporal smoothing to filter out short (< 3 days) events. Specifically, the daily sea
ice concentration values were spatially smoothed using a generic boxcar filter with a square
footprint of 3 x 3 grid cells. The data were then temporally smoothed three times using a Hann
window.
The daily sea ice concentrations are used to define the metrics of the start and end of break-up
and freeze-up in each year of a 40-year period, 1979-2018.  The definitions build on those
used by Johnson and Eicken (2016; hereafter denoted as J&E), which were informed by
Indigenous experts' observations of ice use and ice hazards in coastal Alaska, and relate to
planning and decision-making at the community-scale (Eicken et al., 2014). Here, we expand
the satellite data analysis with minor modifications of the break-up and freeze-up criteria to
broaden the applicability to coastal areas. Examples include imposing maximum and
minimum values for the thresholds computed from summary statistics of the daily sea ice
concentration values of relevant periods. The revised definitions are presented in Table 1 and
the differences relative to those of J&E are listed in Table 2.
The four indicators in this study are the dates of the start and end of break-up and freeze-up.
For purposes of this study, the break-up period may be regarded as the time between the
Arctic sea ice maximum (typically in March) and the sea ice minimum (typically in
September, with June representative of the period most rapid break-up). Similarly, the freeze-
up period extends from September through March, with November representative of the
period of most rapid freeze-up. The corresponding indicators used by Bliss et al. (2019) are
the date of opening (defined as the last day on which the ice concentration drops below 80%
before the summer minimum), the date of retreat (defined as the last day the ice concentration
drops below 15% before the summer minimum), the date of advance (defined as the first day
the ice concentration increases above 15% following the final summer minimum) and the date
of closing (defined as the first day the ice concentration increases above 80% following the
final summer minimum). For the comparisons of indicator dates presented in Section 3, we
did not make any modifications to the Bliss et al. (2019) criteria.
While the various thresholds in Table 1 may seem somewhat arbitrary at first glance, they are
based on past sensitivity tests. In particular, the 10% threshold is based on prior work (J&E)
in which sensitivities were explored. The selected thresholds were those that generally
maximized the number of such years across the coastal locations and MASIE regions.

Table 1.  Definition of the start and end of break-up and freeze-up.

Break-up start    The date of the last day for which the previous two weeks' ice concentration

always exceeds a threshold computed as the maximum of (a) the winter

(January-February) average minus two standard deviations and (b) 15%.

Undefined if the average summer sea ice concentration (SIC) is greater than

40% or if the subsequent break-up end is not defined.

Break-up end      The first date after the break-up start date for which the ice concentration

during the following two weeks is less than a threshold computed as the

maximum of (a) the summer (August-September) average plus one standard

deviation and (b) 50%. Undefined if the daily SIC is less than the threshold

for the entire summer or if break-up start is not defined.

Freeze-up start:  The date on which the ice concentration exceeds for the first time a threshold

computed as the maximum of (a) the summer (August-September) average

plus one standard deviation and (b) 15%. Undefined if the daily SIC never

exceeds this threshold, if the mean summer SIC is greater than 25%, or if

subsequent freeze-up end is not defined.

Freeze-up end:    The first date after the freeze-up start date for which the following two

193                      weeks' ice concentration exceeds a threshold computed as the maximum of

(a) the average winter (January-February) ice concentration minus 10% and

(b) 15%, and the minimum of this result and (c) 50%. Undefined if daily SIC

exceeds this threshold for every day of the search period or if freeze-up start

is not defined.

Table 2.  Changes in the indicator definitions relative to Johnson and Eicken (2016), denoted
as "J&E". The symbol "σ" denotes standard deviation; "sic" denotes sea ice concentration.
*Break-up start:*
- minimum sic threshold created at 15% (J&E: last day exceeding Jan-Feb mean minus 2σ)
- undefined if average summer sic > 40% (J&E: no such criterion)
- undefined if subsequent breakup end date not defined (J&E: no such criterion)

*Break-up end:*
- first time sic below threshold for 2 weeks instead of last day below threshold

(J&E: last exceeding larger of Aug-Sep mean or 15%)

- minimum threshold 50% (J&E: minimum threshold of 15%
- undefined if break-up start not defined (J&E: no such criterion)

*Freeze-up start:*
- first day on which sic exceeds Aug-Sep average by 1σ (J&E: same)
- undefined if mean summer sic > 25% (J&R: no such criterion)
- undefined if subsequent freeze-up end not defined (J&E: same)

*Freeze-up end*:
- first time sic above threshold for following 2 weeks instead of first day above threshold

(threshold is Jan-Feb average minus 10%, as in J&E)

- thresholds imposed: Minimum (15%) and maximum (50%) (J&E: no such thresholds)
- undefined if sic always exceeds threshold (J&E: same)
Our evaluation of the coastal indicators includes comparisons of the various dates (break-
up/freeze-up start/end) at nearshore locations with the corresponding metrics for broader areas
of the Arctic Ocean and the subarctic seas. A set of ten locations was selected on the basis of
their geographical distribution and the relevance of local sea ice to uses by communities,
industry, military or other stakeholders.  Examples of local uses include over-ice travel for
access to marine mammals, offshore travel between coastal communities, access of coastal
facilities by commercial vessels, and protection from coastal waves and erosion. The ten
locations are shown in Figure 2 and listed in Table 3, together with their geographic
coordinates.  While there is admittedly some subjectivity in the selection of these sites, our
priorities were (1) a pan-Arctic geographical distribution, thereby expanding the emphasis on
North American locations in past studies (see Discussion in Section 4) and (2) inclusion of
locations with a mix of users affected by sea ice: Indigenous communities, industry, military
and other stakeholders. For each of these locations, several passive microwave grid cells close
to (but not adjacent to) the coastline were selected for calculation of the break-up and freeze-
up metrics. More specifically, the contamination of the passive microwave-derived ice
concentrations by the presence of land in a grid cell required the exclusion of grid cells
containing land.  Therefore, the selected grid cells satisfied the criterion that they were the
cells closest to the coast but centered at least 25 km from the coast.  Figure 2 shows
geographical insets illustrating the proximity of the selected grid cells to the coastline.
With regard to the grid cell selection, we experimented with the grid cell selections at Sabetta
and Utqiagvik. When the grid cell locations were shifted offshore by one pixel at Sabetta, the
mean break-up start and end dates changed by only -0.1 and -1.1 days, respectively; the
corresponding changes in the freeze-up start and end dates were 0.2 and -0.7 days,
respectively.  At Utqiagvik, the offshore shift resulted in an earlier mean break-up start by 3.3
days and a later mean break-up end by 2.9 days.  The earlier break-up start is consistent with
the presence of fast ice at the coast, as discussed in Section 4.  The changes in Utqiagvik's
freeze-up dates were small when the pixels were shifted offshore, where the start of freeze-up
occurred 1.1 days later and the end of freeze-up 1.1 days earlier than closer to the coast.

Table 3.  Near-coastal locations selected for calculation of break-up and freeze-up metrics

| | Sea | Location | Latitude, Longitude | Significance of location |
|---|---|---|---|---|
| 252 | Beaufort Sea | Prudhoe Bay | 70.2N, 148.2W | oil facilities |
| 253 | Chukchi/Beaufort Seas | Utqiaġvik | 71,3N, 156.8W | Indigenous community |
| 254 | Chukchi Sea | Chukchi Sea | 69.6N, 170W | shipping route |
| 255 | Bering Sea | St. Lawrence Island | 65.7N, 168.4W | Indigenous community |
| 256 | East Siberian Sea | Pevek | 69.8N, 170.6E | port, mining facility |
| 257 | Laptev Sea | Tiksi | 71.7N, 72.1E | research site, port |
| 258 | Kara Sea | Sabetta | 71.3N, 72.1E | port, LNG facility |
| 259 | Greenland Sea | Mestersvig | 72.2N, 23.9W | military base |
| 260 | Baffin Bay | Clyde River | 70.3N, 68.3W | Indigenous community |
| 261 | Hudson Bay | Churchill | 58.8N, 94.2W | port, tourism |

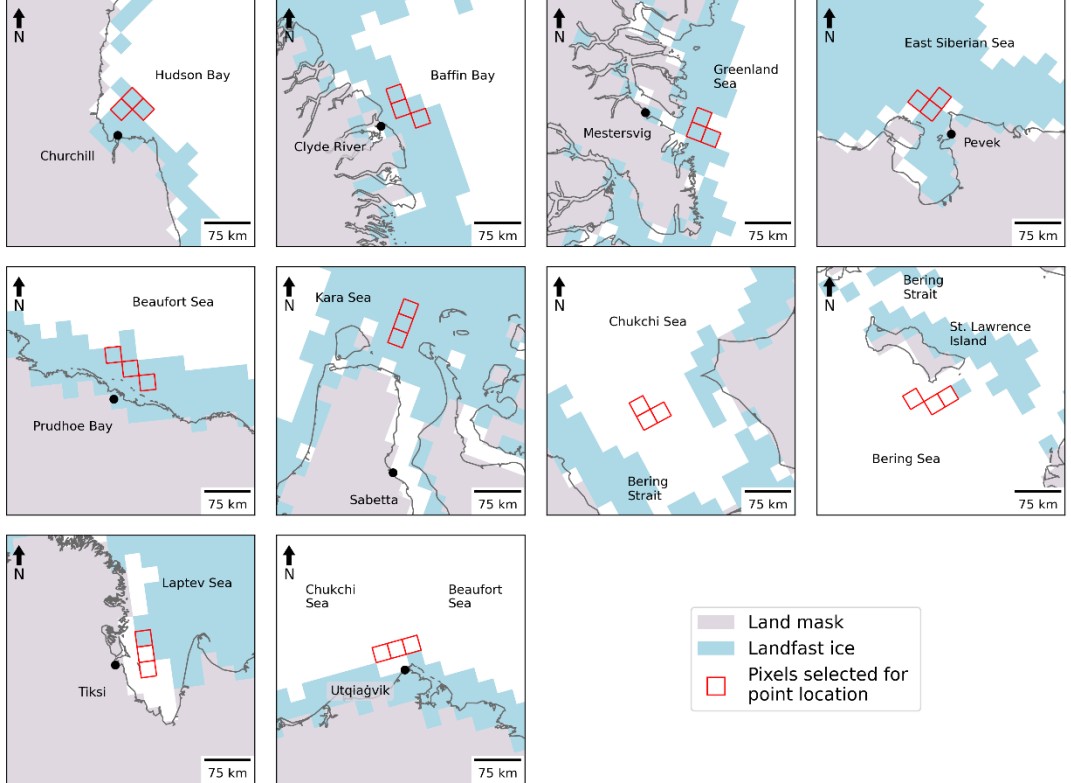

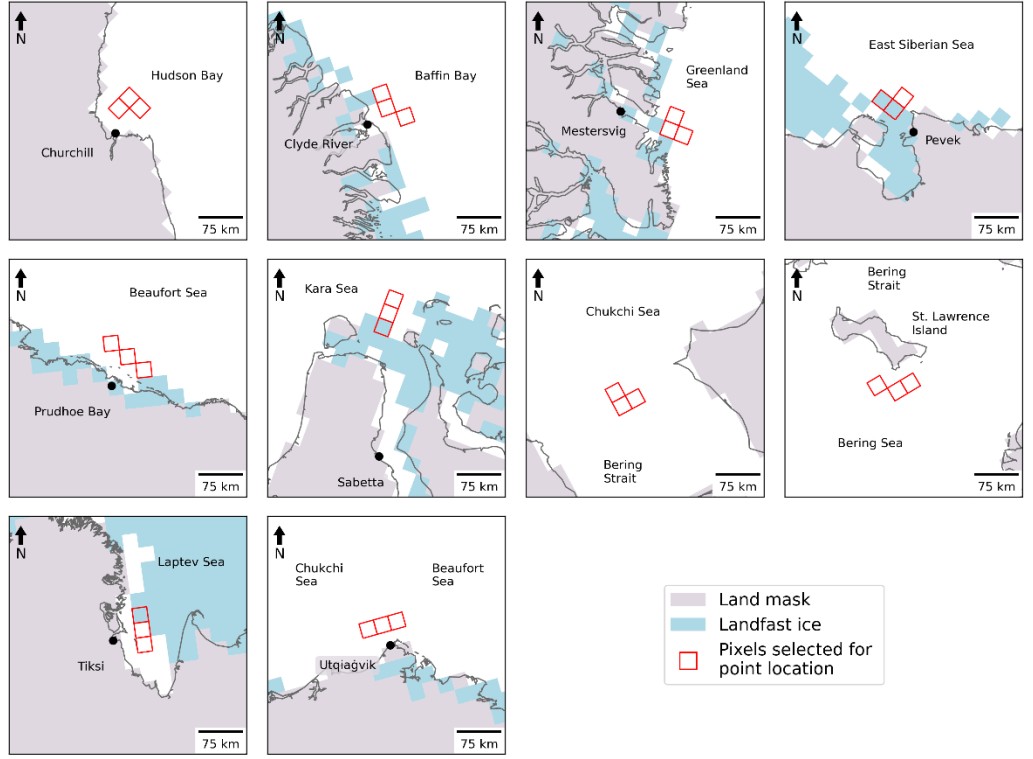

Figure 2. Grid cells (red squares) for which passive-microwave-derived ice concentrations

were used in computing the break-up and freeze-up metrics for the coastal locations. Black

dots represent the actual locations of the coastal communities. Blue shading denotes

maximum (upper panels) and median (lower panels) coverage of landfast ice in June over the

1972-2007 period based on charts of the U.S. National Ice Center --

https://nsidc.org/data/G02172 (accessed 28 June 2022).

It is apparent from Figure 2 that the innermost extent of the landfast ice does not always

coincide with the coastline, which we assume here should always be the inner boundary of

landfast ice. The northern Siberian coast (Sabetta and Tiksi) provides examples. In pursuing

an explanation for the discrepancies, we found that the land mask in the fast ice dataset

(digitized charts of the National Ice Center) differs from the land mask of the NSIDC's

passive microwave dataset. The resulting offset does not change the area covered by sea ice in
each regional plot, but it does result in the mis-location of the inner boundary of landfast ice.
The discrepancy does not alter the reasoning about the geographically varying roles of
landfast ice, as discussed in Section 4, and a more detailed analysis of the origin of these
offsets in coastline depiction and landfast ice location is beyond the scope of this paper.
The grid cell selections for St. Lawrence Island and the Chukchi Sea deserve special
comment. The grid cells off St. Lawrence Island were chosen to reflect timing and location of
subsistence harvests by the communities of Gambell and Savoonga. Because of extensive ice
coverage, including landfast ice, north and northwest of the island, both communities
traditionally conduct bowhead whale harvests at hunting camps on the south side of the island
once spring ice break-up is underway (Noongwook et al., 2007). These sites also reflect the
seasonal migration of whales in waters south of the island with the seasonal retreat of the ice
cover (Noongwook et al., 2007), modulated somewhat by the presence of a polynya south and
southwest of the island (Krupnik et al., 2010; Noongwook et al., 2007). Traditional walrus
harvest practices on St. Lawrence Island await the very end of the bowhead whale hunt
(Kapsch et al., 2010), with timing of spring ice break-up south of the island as the driving
factor. These practices motivated our selection of grid cells southeast of the island. As shown
later (Section 4), landfast ice is confined to the northern coastal region of St. Lawrence Island
– consistent with the frequent presence of the polynya south of the island.  In the case of the
Chukchi Sea, the grid cells are indeed farther from the coast than for the other sites; the
locations were intentionally selected to be farther offshore in order to provide a non-coastal
counter-example to the other sites, all of which are adjacent to a coast.
Previous studies cited earlier have evaluated break-up and freeze-up metrics for subregions of
the Arctic Ocean and the surrounding seas (Markus et al., 2006; Johnson and Eicken, 2016;
Bliss and Anderson, 2018; Peng et al., 2018; Bliss et al., 2019; Smith and Jahn, 2019). For
comparisons with broader regions offshore of our selected sites, we utilize the MASIE
(Multisensor Analyzed Sea Ice Extent) regionalization
(https://nsidc.org/data/masie/browse_regions). Of the MASIE regions shown in Figure 3, we
choose the following for computation of regionally averaged metrics of break-up and freeze-
up: Beaufort Sea, Chukchi Sea, East Siberian Sea, Laptev Sea, Kara Sea, Greenland Sea,
Baffin Bay, Hudson Bay, and Bering Sea.

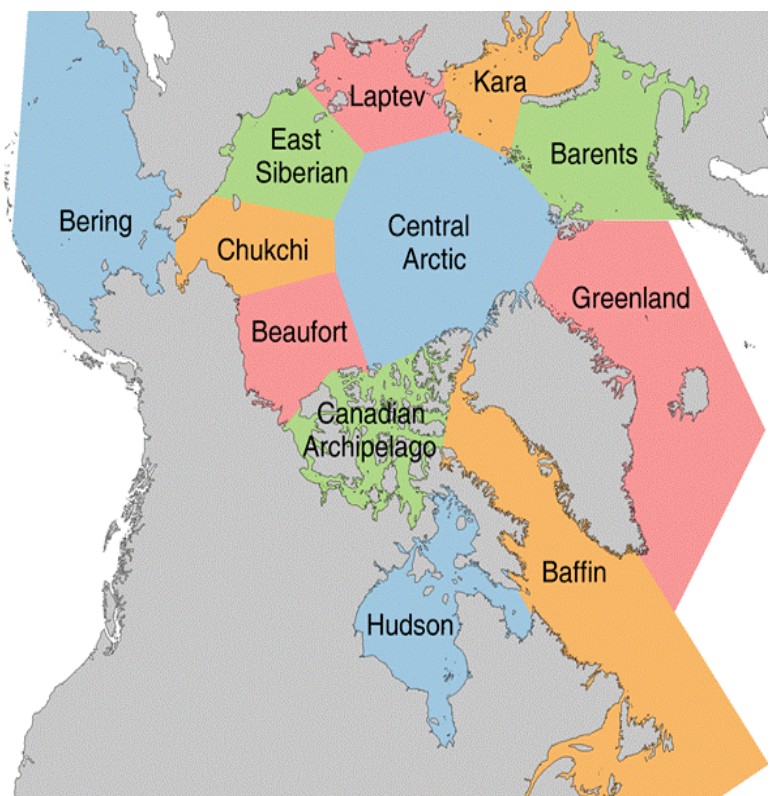


Figure 3.  The MASIE subregions of the Arctic. Regions utilized in this study include
Beaufort Sea, Chukchi Sea, East Siberian Sea, Laptev Sea, Kara Sea, Baffin Bay, Hudson
Bay, and Bering Sea.
The following section includes time series of the local indicators and, for comparison, time
series of the corresponding MASIE regional indicators.  In order to address the spatial
coherence of the indicators, we performed a factor analysis on the different sets (break-
up/freeze-up, start/end dates). The computation of the indicators was dome for the ten local
sites and for the MASIE regions in which they fall.  Factor analysis is a statistical method for
quantifying relationships among a set of variables.  The variability in the overall dataset is
depicted by a set of factors.  Each factor explains a percentage of the total variance in space
and time.  Each variable in each factor is given a loading (or weight) based on its contribution
to the variance explained by that factor.  The first factor can be viewed as the linear
combination of the variables that maximizes the explained variance in the overall dataset.  The
second and each successive factor maximize the variance unexplained by the preceding
factors. Successive factors explain successively smaller fractions of the overall variance.
Multiple variables can have strong loadings in the same factor, indicating they follow a
similar pattern and are likely highly related. Factor analysis has a long history of applications
to Arctic sea ice variability (Walsh and Johnson, 1982; Fang and Wallace, 1994; Deser et al.,
2000; Fu et al., 2021).  The factor analysis calculations used here were performed using the
XLSAT software package run in Excel (https://www.xlstat.com/en/)
**3.   Results**
With coastal ice retreat and onset of ice advance as this study's primary foci, we first
demonstrate the applicability of the indicators evaluated here.   The various metrics of sea ice
break-up and freeze-up in Table 1 are not defined for all locations in the Arctic.  For example,
locations that remain ice-covered throughout a particular year will not be assigned dates for
any of the indicators in that year, and the same is true of locations at which sea ice does not
form during a particular year. Figure 4 shows the number of years in the 1979-2018 study
period during which the break-up and freeze-up indicators are actually defined.  It is apparent
that the indicators are consistently defined in the seasonal sea ice zone spanning the subarctic
seas. In particular, all ten coastal locations in Table 2 are in the yellow areas (>35 years out of
40 years defined) of Figure 4. Of note in Figure 4 is that the number of years with defined
break-up indicators slightly exceeds (by one) the number of years with freeze-up indicators at
some locations at the outer periphery of the seasonal sea ice zone.  These are locations in
which sea ice was present for some portion of the early years but not at the end of the study
period, so in one of the years there was a break-up but no freeze-up.

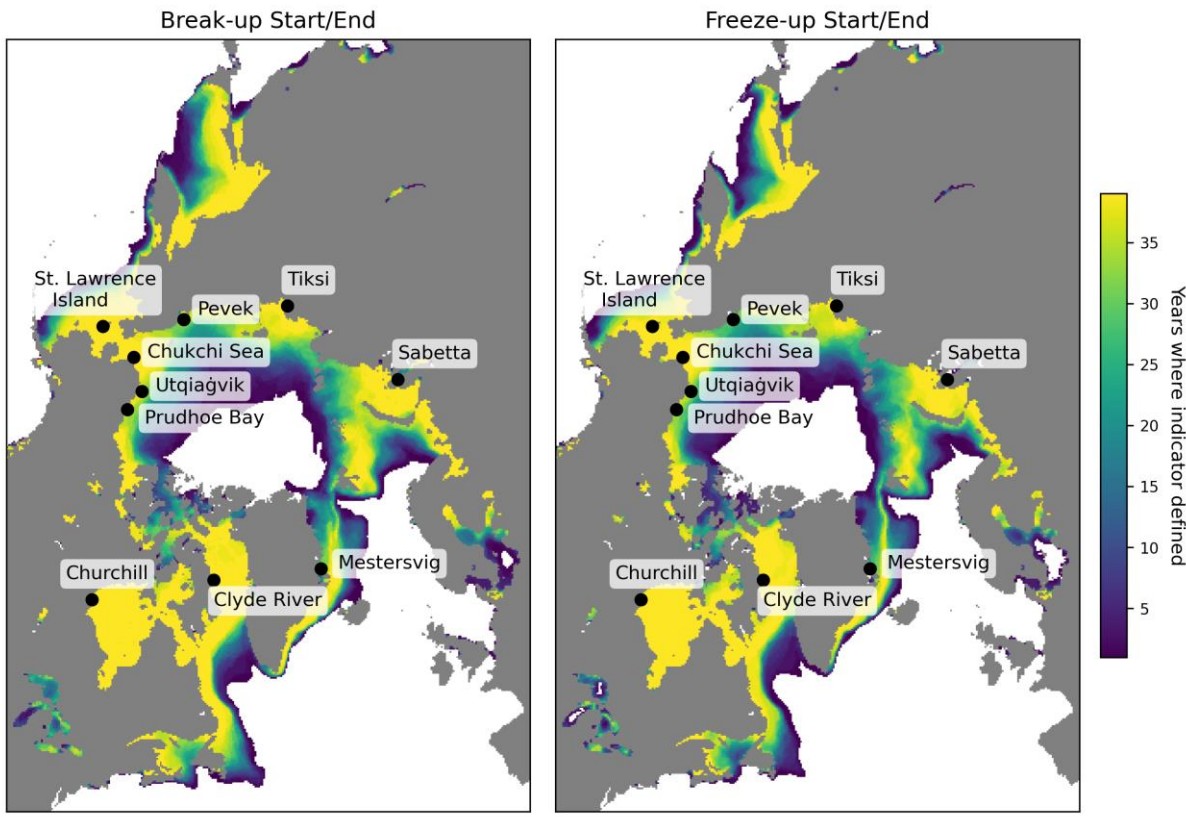


Figure 4.  Number of years in the 1979-2018 study period in which the break-up and freeze-up
indicators were defined. Note that end dates for break-up and freeze-up exist only for years in
which there are start dates for break-up and freeze-up.  The start and end dates of the overall
data record (1 Jan 1979 – 31 Dec 2018) can result in differences of 1 year in the counts when
freeze-up occurs around January 1.

A key issue to be addressed is the degree to which the indicators utilized here differ from
those of previous studies. The metrics of Bliss et al. (2019) or similar variants have been used
in recent publications and provide natural points of comparison.  While there are various
differences between our metrics and those of Bliss et al., the most consequential for the
computed dates is the use of departures from winter/summer averages concentrations in our
criteria vs. Bliss et al.'s use of 15% and 80% concentrations as key thresholds. This
distinction is analogous to the difference between the NASA Team algorithm's use of fixed
tie points and the NASA Bootstrap algorithm's use of "dynamic" (time/space-varying) tie
points.
Figure 5 and Table S1 show that there are systematic differences between our metrics (based
on the modified J&E criteria) and those of Bliss et al. when the two sets of metrics are
evaluated for the MASIE regions.  In particular, J&E's start and end of breakup generally
occur earlier by up to several weeks than the corresponding dates of opening and retreat
defined by Bliss et al.  On the other hand, J&E's freeze-up dates are more closely aligned with
those of Bliss et al., although J&E's end-of-freeze-up occurs later (by 1 to 3 weeks) than Bliss
et al.'s closing date in most of the MASIE regions, especially the North Atlantic and Canadian
regions.

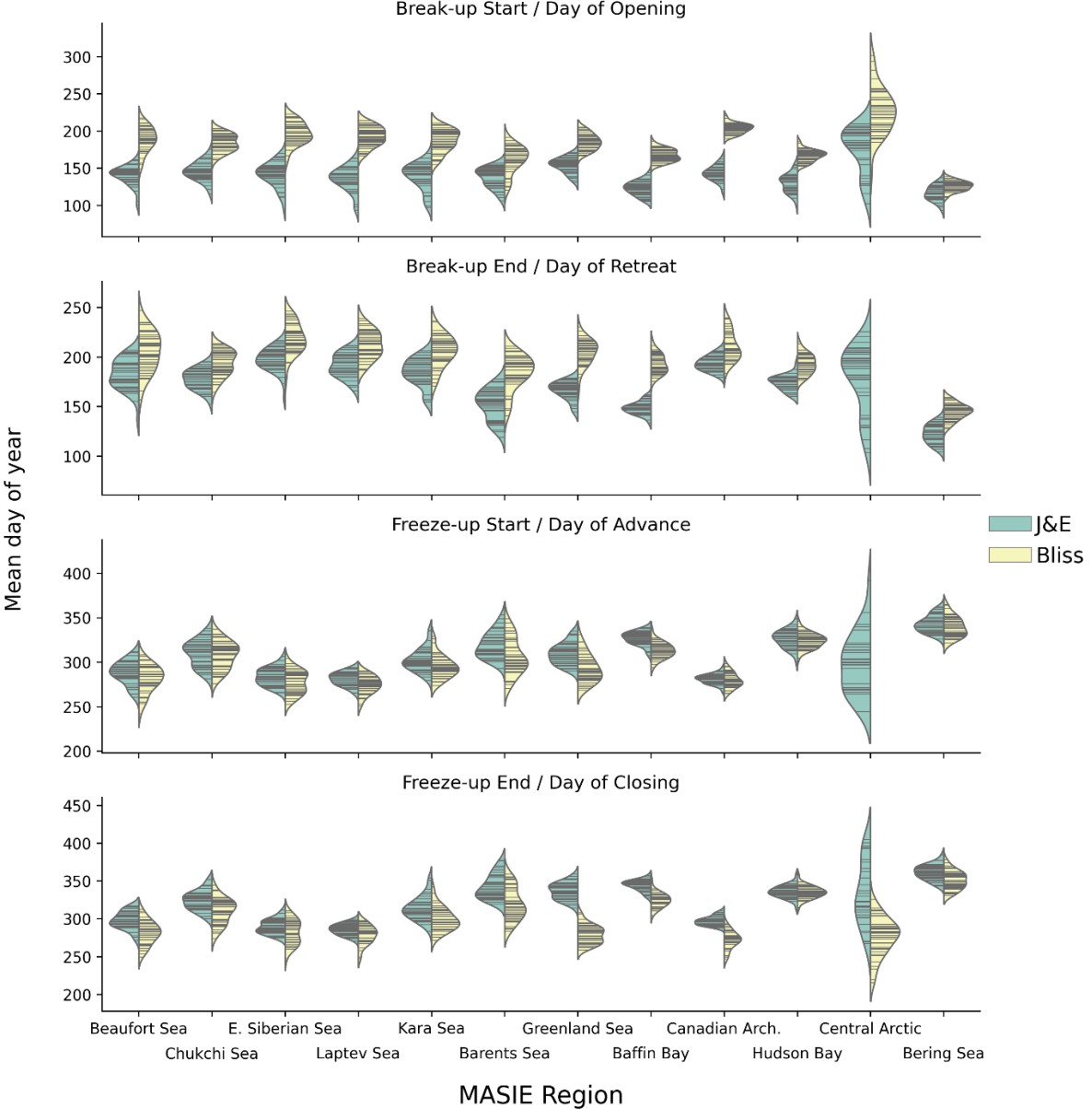

Figure 5. "Violin" plots of the Julian dates of the break-up/freeze-up metrics used in this study based on Johnson and Eicken (2016) (green shading) and the corresponding dates of ice opening, retreat, advance and closing as defined by Bliss et al. (2019) (yellow shading). A violin plot shows a distribution by widening the horizontal lines in the ranges (of day of the year, in this case) having the highest concentration of values. The thin black lines represent

the observations themselves; the black strips are clusters of lines representing groups of
similar values in the distribution. The violin plots provide no information about the temporal
sequence of the values.
The violin plots in Figure 5 show distributions but not the temporal variations that have been
indicated by results of previous studies (Peng et al., 2018; Bliss et al., 2019). Figures 6 and 7
provide the temporal perspective on the end dates of break-up (Day of retreat) and freeze-up
(Day of closing), respectively.  In each of the MASIE regions, the J&E criterion gives an
earlier break-up date.  The difference is typically two to three weeks, although it exceeds a
month in the Greenland Sea and Baffin Bay.  Despite the offsets, the trends are nearly the
same in nearly all the regions.  Exceptions are the Canadian Archipelago, where the J&E trend
is weaker than the Bliss trend, and the Bering Sea, where the trends are opposite in sign.
However, the trend in the Bering region is not statistically significant at the 99% level by
either metric, in contrast to all other regions in which the trends are significant at this level
(Table S2).  The main conclusion from Figure 6 is that, except for the Bering Sea, sea ice
break-up is occurring earlier throughout the Arctic than several decades ago, no matter which
metric is used.
In contrast to the trends towards earlier breakup, the J&E and Bliss metrics for the end of
freeze-up both show significant trends towards later dates in most of the MASIE regions
(Figure 7 and Table S3). In this case, even the Bering Sea shows a trend towards later freeze-
up.  Again, there is an offset towards a later date with the J&E metric, although the offset has
a range among the regions, from essentially zero in Hudson Bay to more than six weeks in the
Greenland Sea.  The trends, however, show less agreement in some regions than do the trends
for break-up dates in Figure 6.  The J&E trends are less positive than the Bliss trends in the

seas of the eastern Russian sector: the Chukchi, East Siberian and Laptev Seas. The same is true, although to a lesser degree, in the Barents Sea and the Canadian Archipelago. The main message from Figure 7 is that the freeze-up is ending later throughout the Arctic, although the magnitude of the trend is more sensitive to the criteria used for end-of-freeze-up than for end-of-break-up.

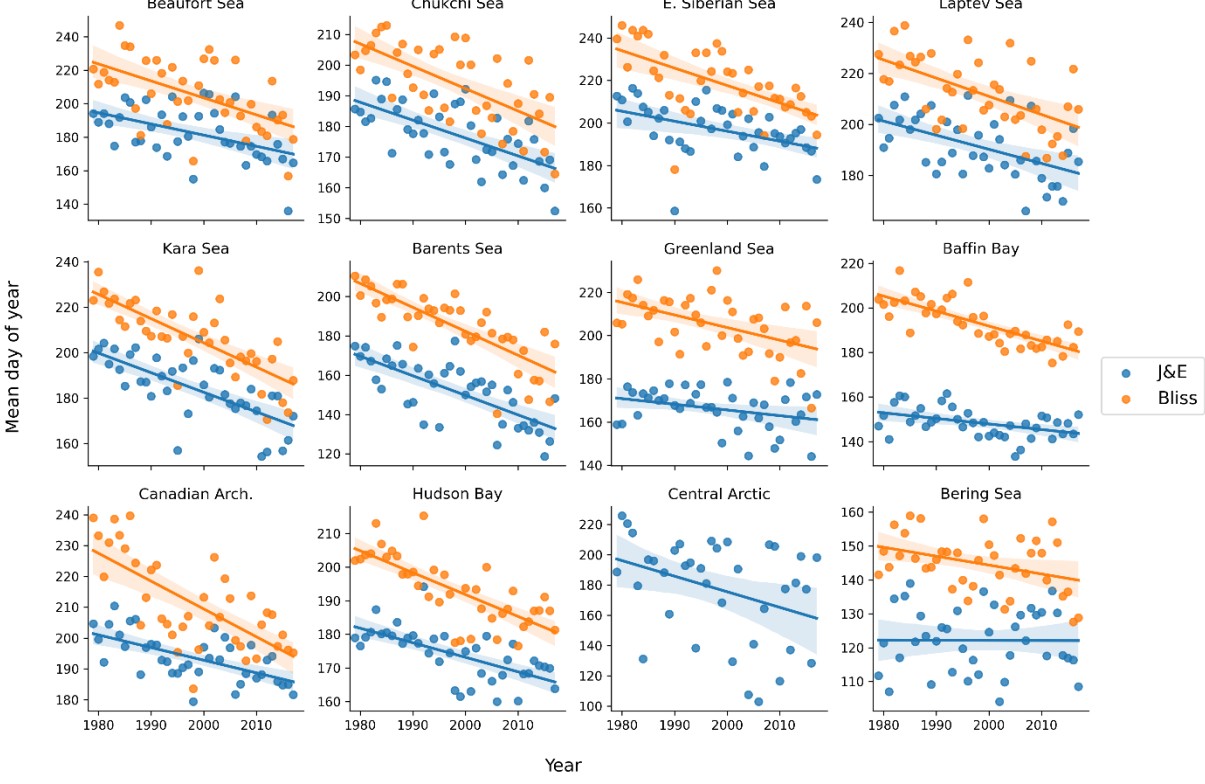

Figure 6. Yearly values of J&E's break-up end date (blue symbols) and the Bliss et al.'s (2019) Day of retreat (orange symbols) in the various MASIE regions. Corresponding trend lines are shown in each panel. (For the Central Arctic region, Bliss et al.'s "Day of retreat" metric is not shown because it was defined for fewer than half the years). Y-axis labels represent day of the year. Date scales on y-axis vary among panels in order to optimize display of data points. For numerical values of slopes and significance levels, see Table S2.

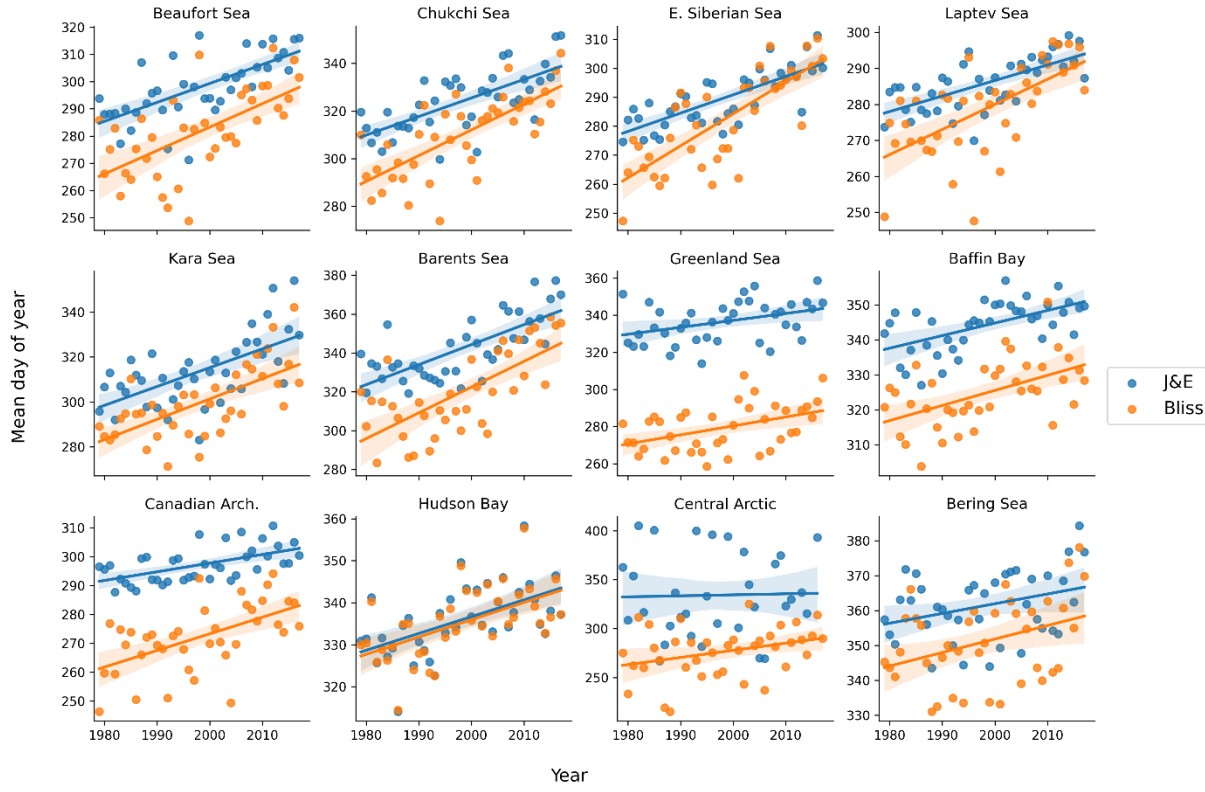

Figure 7.  Yearly values of J&E's freeze-up end date (blue symbols) and the Bliss et al.'s

(2019) Day of closing (orange symbols) in the various MASIE regions. Corresponding trend

lines are shown in each panel. Y-axes labels represent day of the year. Date scales on y-axis

vary among panels in order to optimize display of data points. Numerical values of slopes and

their significance levels are provided in Table S3.

A final comparison is presented in Figure 8, which shows the ice season lengths computed

using the two sets of metrics.  The ice season length is defined as the number of days between

the end of freeze-up and the start of break-up.  Consistent with J&E's earlier break-up (Figure

6) and later freeze-up (Figure 7), the J&E metrics yield a shorter ice season than the Bliss et al

metrics.  The differences in Figure 8 exceed a month in most of the Arctic except for the

Bering Sea, Hudson Bay and the Canadian Archipelago.  However, the negative trends of ice
season length are similar in magnitude according to both sets of metrics over most of the
Arctic. The trend maps are not shown here because they add little to the information conveyed
in Figures 6 and 7.

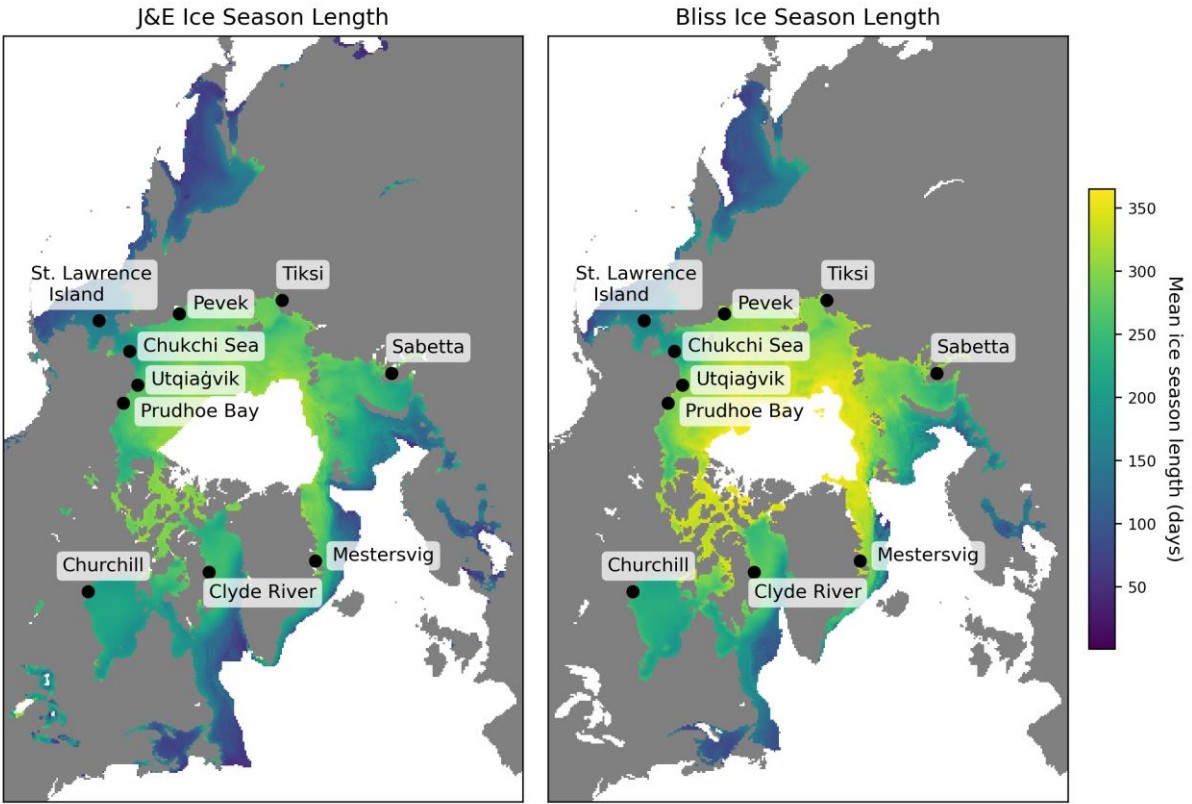


Figure 8.  Mean ice season length based on the J&E metrics (left) and the Bliss et al. (2019)
metrics (right).  Metrics of break-up and freeze-up were not defined in a sufficient number of
years in the white area near the North Pole.
Given that this study targets the use of local indicators, it is important to assess the
relationship between the local indicators and those for the broader MASIE regions containing
the coastal locations.  An important caveat in such a comparison is that our local indicators
were designed for coastal users, not for broader regional or applications in areas far from
shore. This distinction introduces the possibility that the coastal indicators may be less than
optimal for the larger MASIE regions.  Figures 9-10 provide these comparisons for the break-
up metrics defined by the modified J&E algorithms.  In all cases, the yearly values (and linear
trend lines) for the ten coastal locations in Table 3 are plotted for the 1979-2018 period,
together with the values for the corresponding MASIE regions.
The break-up start dates (Figure 9) differ between the coastal locations and the broader
MASIE regions in most of the ten cases, and in some cases the trends are notably different.
With regard to systematic differences, not only the magnitude but also the sign of the offsets
varies among the regions.  The break-up start date at the coast is later than for the MASIE
regions for Prudhoe (Beaufort Sea), Utqiaġvik (Chukchi Sea), Tiksi (Laptev Sea), and both
Canadian locations: Churchill (Hudson Bay) and Clyde River (Baffin Bay).  These sites are all
Arctic coastal locations at which varying extents of landfast ice are present.  By contrast, the
coastal locations have earlier break-up start dates (relative to their corresponding MASIE
regions) at St. Lawrence Island, Mestersvig (Greenland Sea) and the Bering Strait (Chukchi
Sea.  The relation of landfast ice to the timing of break-up is discussed further in Section 4.
While the general trend towards earlier break-up noted above (Figure 6) is apparent at most of
the coastal locations, the magnitudes of the trends can differ between the coastal sites and the
broader MASIE regions.  Figure 9 shows that the trend towards an earlier start of break-up is
stronger at the coastal location relative to the MASIE region at Churchill, Clyde River, Pevek
and Sabetta. Only at Tiksi is the negative trend weaker at the coastal site. In the other regions
the trends are nearly identical.

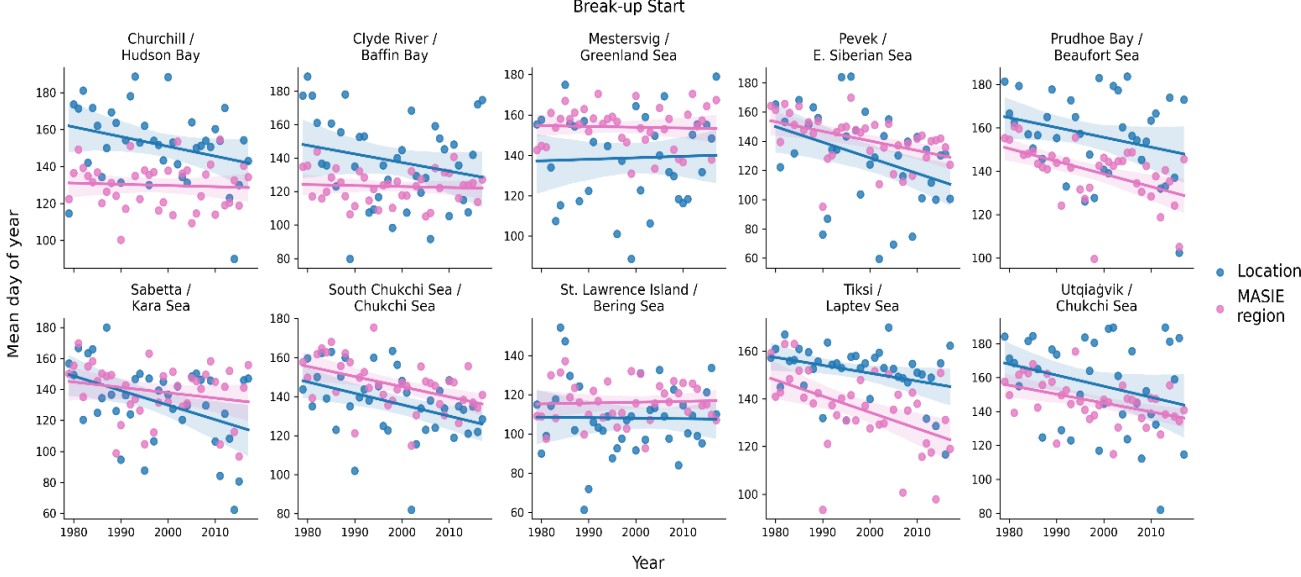

Break-up Start


Figure 9.  Yearly values (1979-2018) of the break-up start dates (shown as day-of-the-year numbers)
for the coastal locations (blue) and the corresponding MASIE regions (pink). Date scales on y-axis vary
among panels in order to optimize display of data points. Linear regression lines are shown with the
same color coding. In each panel, the upper line of header identifies the coastal location and the lower
line identifies the MASIE region. All values are based on the modified J&E algorithms. Slopes and
their significance levels are listed in Tables S2 and S3.
The break-up end dates (Figure 10) show differences similar to those in Figure 9 in most, but
not all, cases.  The break-up end date occurs later at Clyde River, Prudhoe and Utqiagvik
relative to the MASIE regions, as is the case with the results in Figure 9.  However, unlike the
break-up start date, the break-up end date also occurs latr at Mestersvig than for the Greenland
Sea MASIE region. The opposite relationship is found in the Kara Sea / Sabetta and the
Chukchi Sea (Bering Strait), where the MASIE region has the earlier break-up end date. The
temporal trends in the break-up end dates are generally similar for the coastal locations and
the MASIE regions, and there are no differences in sign.  All coastal locations and all MASIE
regions show negative trends, i.e., trends toward earlier break-up end dates in recent decades.

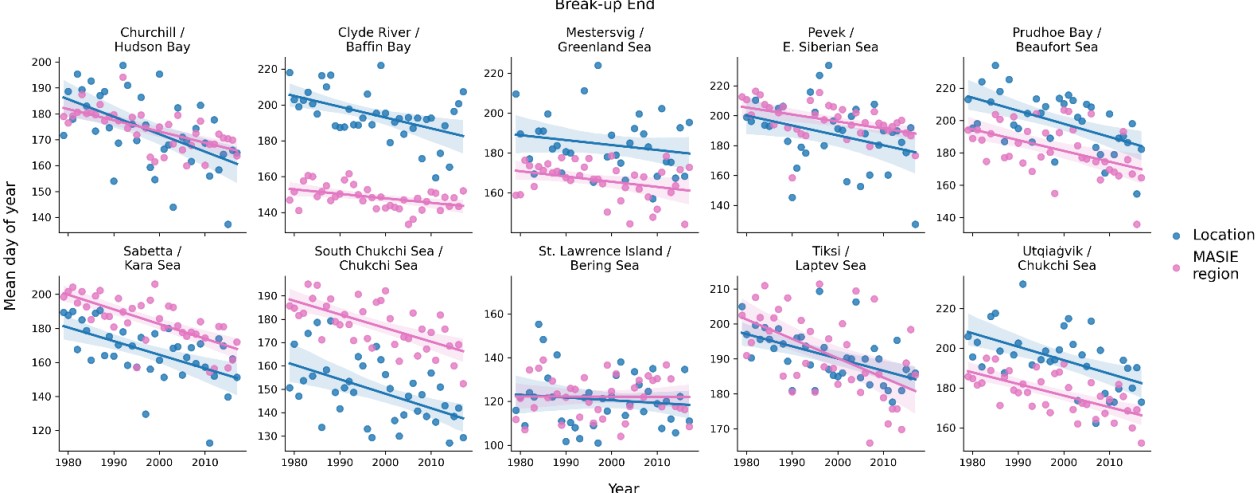


Figure 10.  Yearly values (1979-2018) of the break-up end dates (shown as day-of-the-year

numbers) for the coastal locations (blue) and the corresponding MASIE regions (pink). Date

scales on y-axis vary among panels in order to optimize display of data points. Linear

regression lines are shown with the same color coding. In each panel, the upper line of header

identifies the coastal location and the lower line identifies the MASIE region. All values are

based on the modified J&E algorithms. Slopes and significance levels are listed in Tables S2

and S3.

The freeze-up start dates are compared in Figure 11. Several regions show large offsets, most

notably Clyde River (Baffin Bay) and Mestersvig (Greenland Sea), where the start of freeze-

up occurs earlier at the coast by several weeks. Both Baffin Bay and the Greenland Sea are

large MASIE regions (Figure 2), favoring the delay of freeze-up start over a substantial

portion of the seasonal sea ice zone within the respective MASIE regions. Freeze-up start

dates are also earlier than offshore at several other coastal locations: Churchill, Sabetta and

Utqiaġvik.  These are regions in which it is common for ice to form along the coast in autumn,

with the ice edge advancing offshore to meet the expanding main ice pack as freeze-up

progresses. Figure 12 shows examples of this dual advance of the freeze-up "front" along the

coasts of the East Siberian Sea in 2021 and the Beaufort Sea in 2020 and 2021. By contrast,

the southern Chukchi Sea location has a later freeze-up date than the Chukchi MASIE region,

largely because the southern Chukchi grid cells are located in an area of relatively warm

inflowing currents from the Bering Sea and are in the southern portion of the Chukchi MASIE

region. As with the break-up end dates, all coastal locations and MASIE regions show trends

of the same sign. In this case, the trends are all positive, indicating a later start to freeze-up.

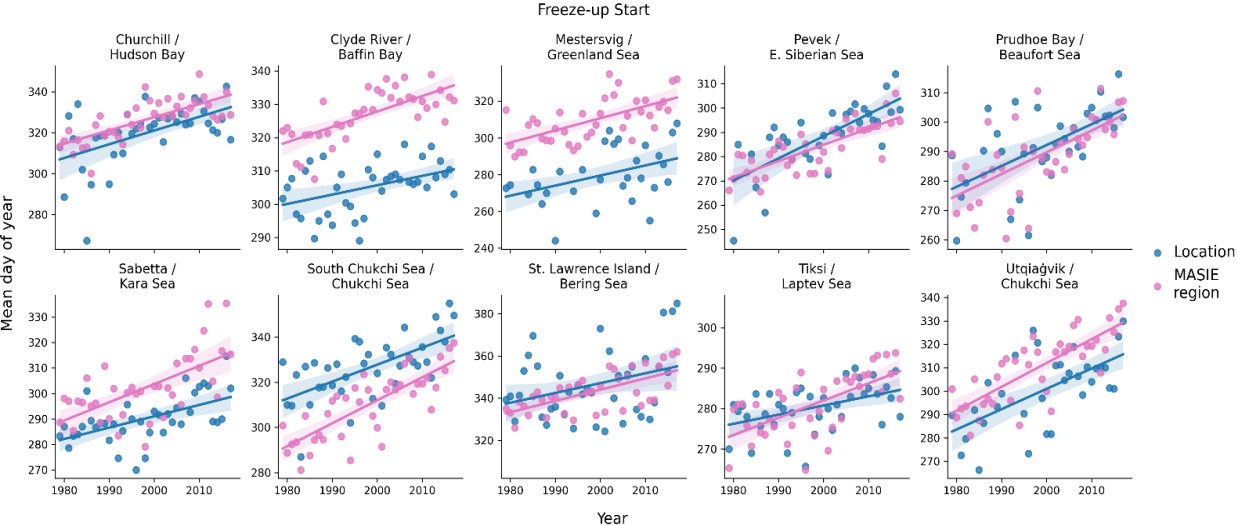

Figure 11. Yearly values (1979-2018) of the freeze-up start dates (shown as day-of-the-year

numbers) for the coastal locations (blue) and the corresponding MASIE regions (pink). Date

scales on y-axis vary among panels in order to optimize display of data points. Linear

regression lines are shown with the same color coding. In each panel, the upper line of header

identifies the coastal location and the lower line lists the MASIE region. All values are based

on the modified J&E algorithms. See Tables S2 and S3 for slopes and significance levels.

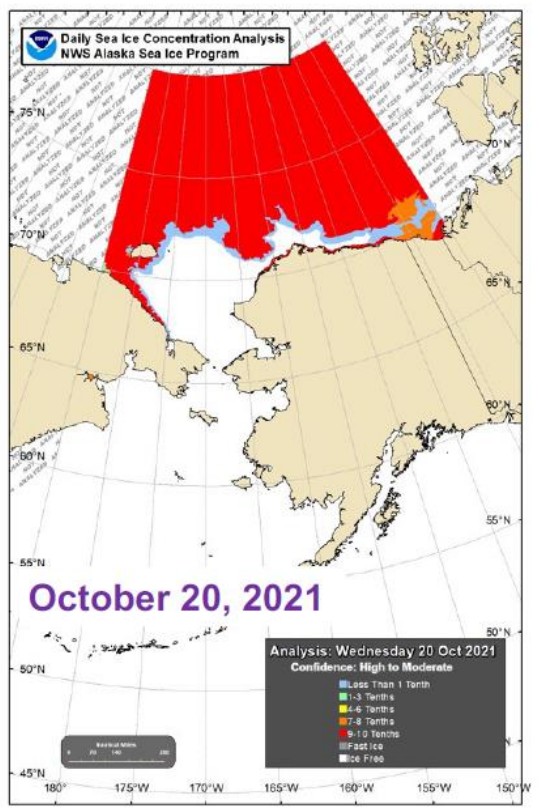 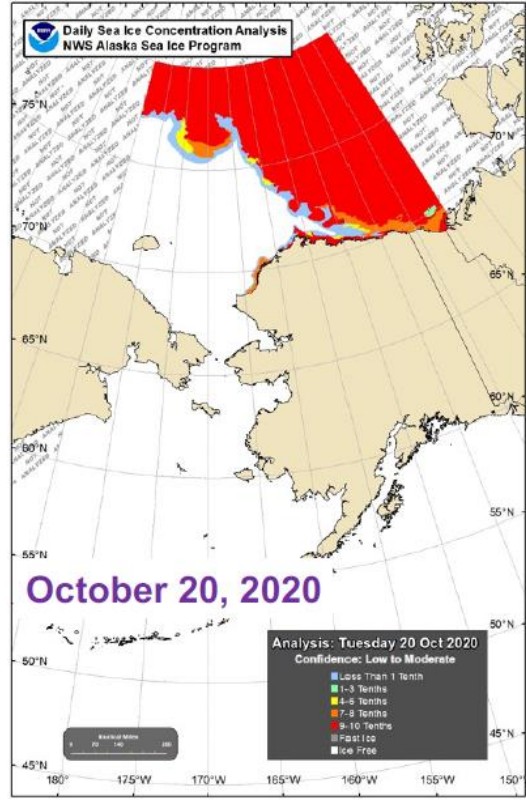

Figure 12. Sea ice coverage on October 20, 2021 (left panel) and October 20, 2020 (right

panel). As indicated by legends in lower right of each panel, red denotes essentially complete

ice coverage, while gray areas have low concentrations. Source: NWS Alaska Region Sea

Ice Desk.

Finally, Figure 13 compares the freeze-up end dates for the ten coastal sites and their MASIE

regions. The results are quite similar to those for the freeze-up start dates in Figure 11.

Relative to the MASIE regions as a whole, freeze-up ends earlier at both Canadian sites

(Churchill and Clyde River), Mestersvig, Sabetta and Utqiaġvik. Again, the differences are

especially large (more than a month) at Clyde River and Mestersvig, both of which are in

large MASIE regions as noted above. The southern Chukchi Sea and, to a lesser extent in

recent decades, Pevek (East Siberian Sea) show later freeze-ups near the coast than for the
MASIE region. Once again, all trends are positive, pointing to a later end to freeze-up at
coastal as well as offshore regions throughout the Arctic. The changes in the freeze-up dates
over the 40-year period are especially large, exceeding one month, at Pevek (East Siberian
Sea) and Prudhoe (Beaufort Sea). The changes are close to a month at Utqiaġvik (Chukchi
Sea) and the Southern Chukchi Sea.

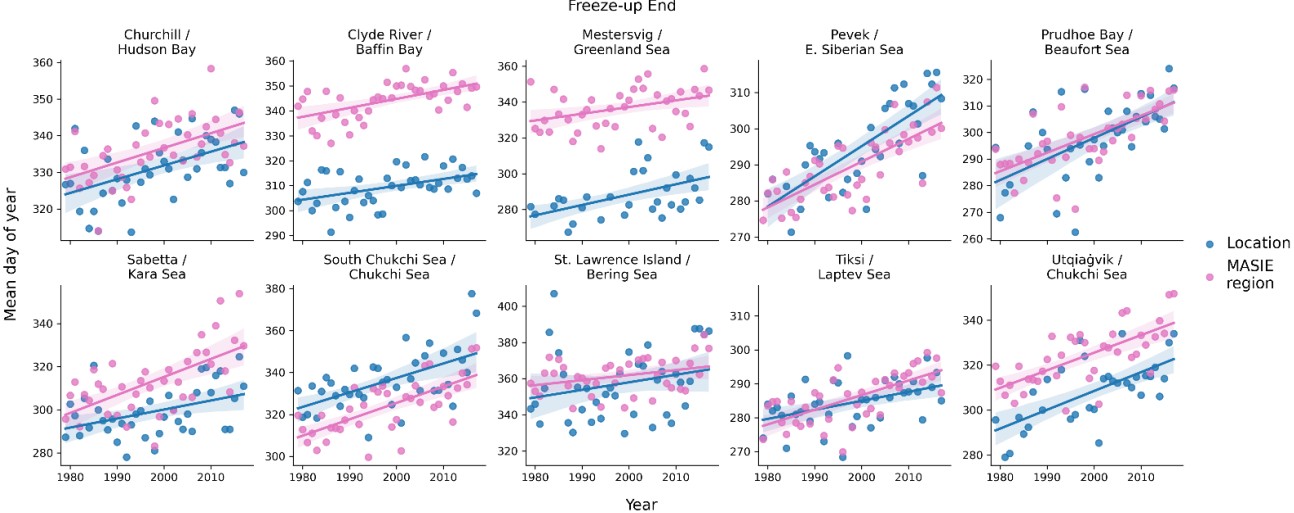


Figure 13. Yearly values (1979-2018) of the freeze-up dates (shown as day-of-the-year
numbers) for the coastal locations (blue) and the corresponding MASIE regions (pink). Date
scales on y-axis vary among panels in order to optimize display of data points. Linear
regression lines are shown with the same color coding. In each panel, the upper line of header
identifies the coastal location and the lower line identifies the MASIE region. All values are
based on the modified J&E algorithms. Slopes and their significance levels are listed in Tables
S2 and S3.
In order to synthesize the information provided by the local indicators, we applied a factor
analysis to each of the four local indicators described in Section 2. For the local indicators,
each input matrix was 10 (locations) x 40 (years).  For comparison, we also applied the factor
analysis to the corresponding regional sea ice areas from the MASIE database (National Snow
and Ice Data Center dataset G02135_v3.0-4).  Because the Chukchi Sea is the MASIE region
for two of the local indicators (Chukchi Sea and Utqiaġvik), the data matrix for the MASIE
regional factor analysis contained 9 (regions) x 40 (years) entries.  We performed the MASIE
factors separately for middle months of the break-up and freeze-up seasons (June and
November, respectively).
In all cases, the first factor contains loadings of the same sign for all locations/regions and is
essentially a depiction of the temporal trends, which account for substantial percentages of the
variance.  The second factor consists of loadings of both signs, corresponding to positive
departures from the mean at some locations and negative departures at others. Figure 14
illustrates this behavior for (a) the break-up start dates and (b) the freeze-up end dates. While
every one of the ten locations has a positive loading in Factor 1, the mixed signs of the Factor
2 loadings point to a regional clustering of the dates.  For example, Figure 14a shows that the
northern coastal sites in the Pacific hemisphere from 90°E eastward to 90°W (Prudhoe Bay,
Utqiagvik, Tiksi, Pevek) have a component of break-up start date variability that is out of
phase with the locations in the western Atlantic/eastern Canada sector from 90°W eastward to
90°E (Mestersvig, Churchill, Clyde River).
The interpretation of Factor 1 as a trend mode is supported by Figure 15, which shows the
time series of the scores of Factor 1 for (a) the break-up start date and (b) freeze-up end dates.
The trends towards an earlier start of break-up and a later end of freeze-up are clearly evident.
Figure 15 also illustrates the tendency for occasional "outlier" years to be followed by a
recovery in the following year.  These plots and those for the other local indicators show that
these extreme excursions and recoveries are superimposed on the strong underlying trends,
resulting in new extremes when the sign of an extreme year is the same as the sign of the
underlying trend.

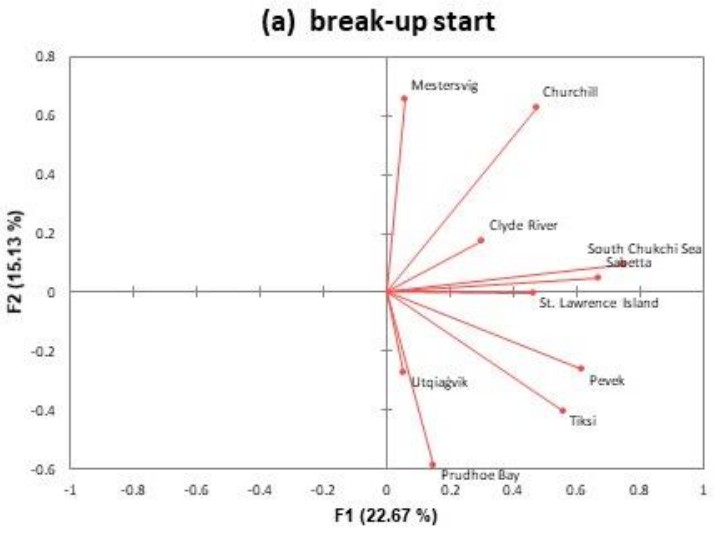

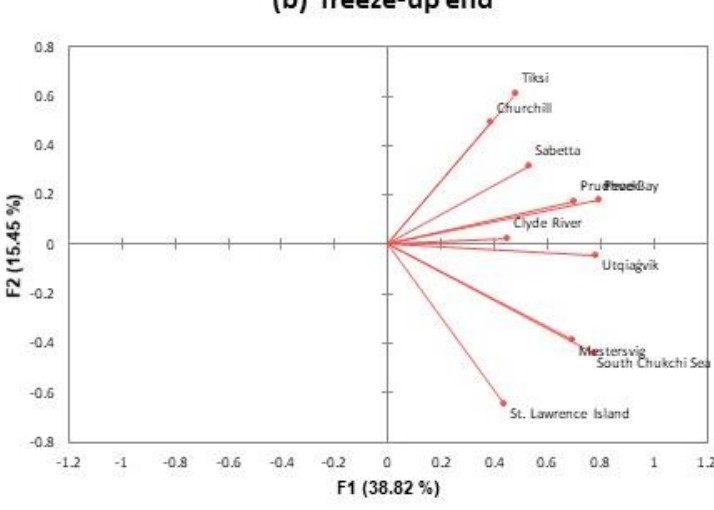


Figure 14.  Loadings for Factor 1 (x-axis) and Factor 2 (y-axis) for (a) the start of break-up and (b)
the end of freeze-up at the ten local coastal sites.  Labels on vectors denote locations.

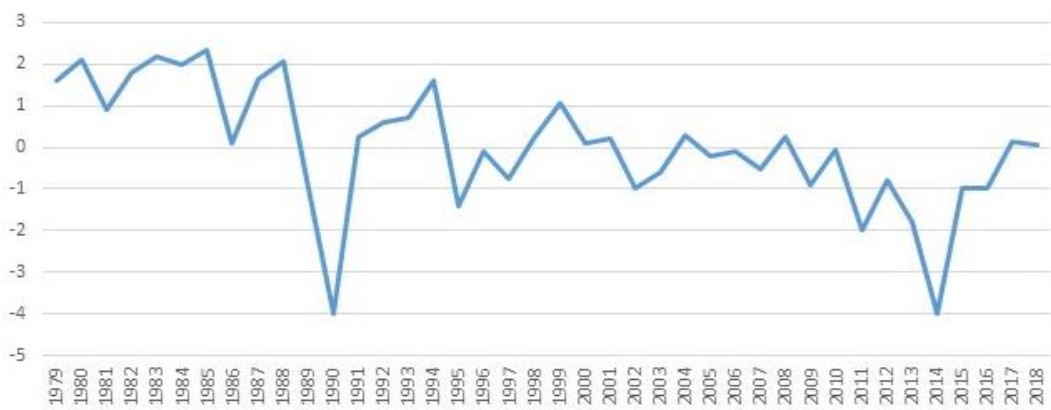

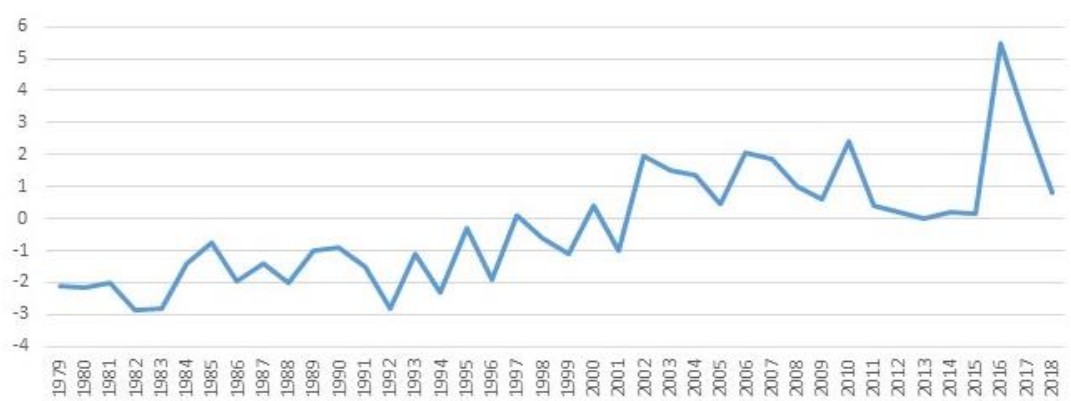


Figure 15.  Scores (time series) for Factor 1 of (a) the start of break-up and (b) the end of
freeze-up at the ten local coastal sites.
Table 4 shows that the first two factors explained more than half the variance for all local and
MASIE indicators except the local break-up start date. The break-up start date is notable for
the small percentages of variance explained by the first two factors.  The implication is that
local conditions play a relatively greater role in the timing of the start of break-up. These local
factors can include landfast ice, inflow of water and heat from the adjacent land areas
(including rivers), and possibly other effects related to local ocean currents or local weather
conditions.  The freeze-up start date has the most spatial coherence in the trend mode (55.7%
of the explained variance).  However, as shown by the last two lines of Table 4, the MASIE
regional ice areas have even greater percentages of variance explained by the first two factors.
In both the break-up and freeze-up seasons (June and November), the first two factors explain
more than 60% of the variance (vs. 37.8%-55.7% for the local indicators).  Because the
variance of the ice concentrations in the MASIE regions is generally greater in the southern
compared to the northern portion of the region, factors for individual MASIE regions have
greater loadings in the south.  However, this does not provide an obvious explanation for why
the percentage of variance explained by the first factor is greater for the MASIE indicators
than for the local indicators. These differences again point to the importance of local
conditions relative to the broader underlying trend in ice coverage, as Factor 1 (the trend)
accounts for most of the differences between the local and regional results in Table 4.

Table 4.  Percentages of variance explained by Factors 1 and 2. Numbers in parentheses are
the contributions of the individual factors (Factor 1 + Factor 2).

| | | | |
|---|---|---|---|
| 585 | Break-up start (local) | 37.8% | (22.7% + 15.1%) |
| 586 | Break-up end (local) | 50.9% | (37.6% + 13.3%) |
| 587 | Freeze-up start (local) | 55.7% | (40.1% + 15.6%) |
| 588 | Freeze-up end (local) | 54.3% | (38.8% + 15.5%) |
| 589 | | | |
| 590 | MASIE ice areas: June | 60.9% | (47.1% + 13.8%) |
| 591 | MASIE ice areas: November | 64.1% | (48.7% + 15.4%) |


Finally, Figure 16 illustrates the tendency for tighter clustering in the regional indicators. For
both the June and November results, the clustering in Figure 16 is clearly more distinct than in
Figure 14, which is the corresponding figure for the local indicators.  The clustering in Figure
16 is geographically coherent, e.g., the Pacific sector sites (Bering, Chukchi, East Siberian)
are in a distinct cluster for the June (break-up), while subclusters for November include the
Hudson and Baffin regions, the Kara and Laptev regions, and the Bering and Chukchi regions.
The results imply that underlying trends and spatially coherent patterns of forcing will be
more useful in explaining – and ultimately predicting – variations of regional sea ice cover.
However, diagnosis and prediction of local indicators will require a greater reliance on
additional information such as local geography and local knowledge, including information
from residents and other stakeholders who have had experience with break-up and freeze-up
of sea ice in the immediate area.


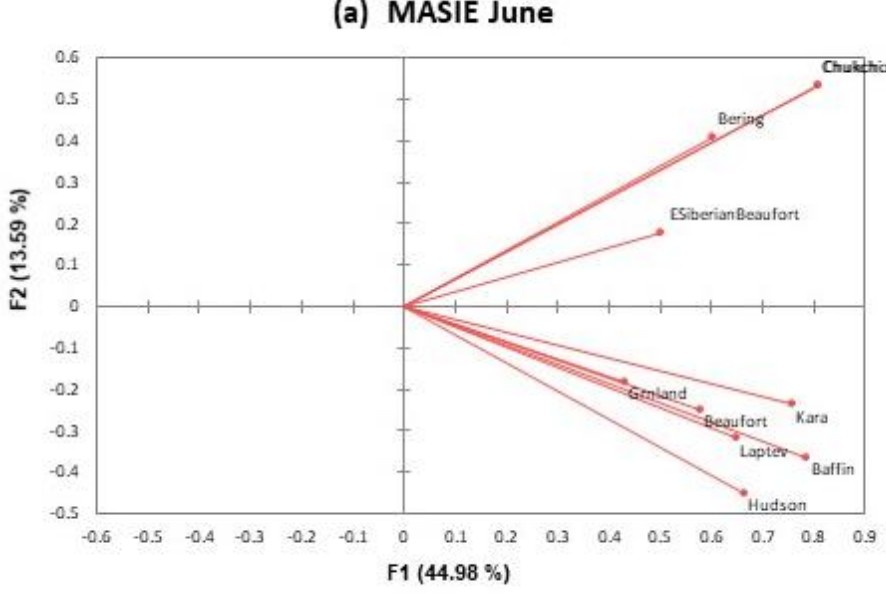

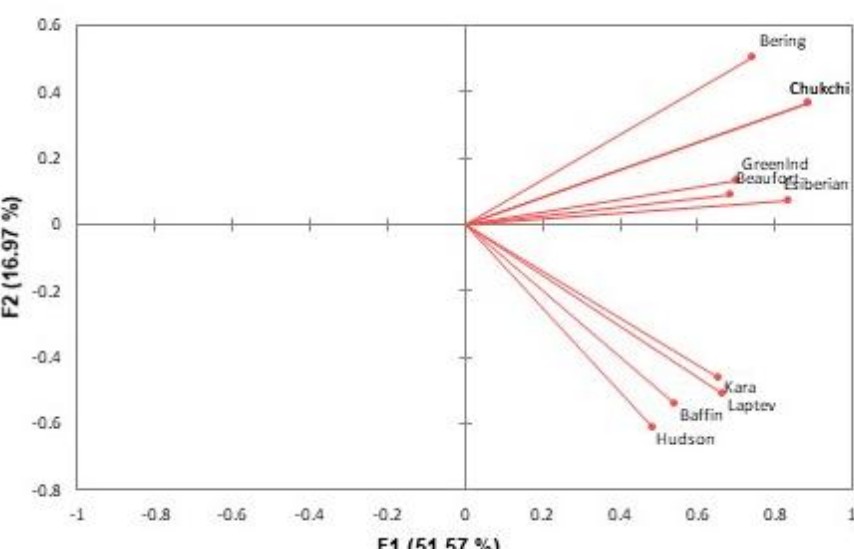


Figure 16.  Loadings for Factor 1 (x-axis) and Factor 2 (y-axis) for the MASIE regional ice

areas of (a) June and (b) November.   Labels on vectors denote MASIE regiona.


**4.  Discussion**
The results presented in Section 3 point to a lengthening of the open water season as a result
of both an earlier break-up and a later freeze-up. The timing of break-up and freeze-up differs
between the coastal sites and the broader MASIE regions that are centered farther from shore
than the coastal grid cells. These differences can be related to the presence of landfast ice,
which characterizes the nearshore coastal waters to varying degrees at most of our coastal
sites (Figure 1).
Landfast ice generally persists longer than pack ice in the adjacent offshore in spring. This
contrast can be explained largely in terms of the stationary nature of the landfast ice cover,
with grounded pressure ridges and confinement by coastal barrier islands (e.g., in the Beaufort
and Kara Seas) locking the ice cover in place. Differences in ice thickness, with offshore sea
ice younger and hence thinner in areas of coastal polynyas with winter new-ice formation
(e.g., in the Chukchi, Beaufort and Laptev Seas) may also contribute to longer persistence of
landfast ice. Finally, with thermal decay of sea ice as a key break-up mode, the absorption of
solar shortwave energy in leads and openings in the offshore ice pack promotes thinning and
decay of the offshore ice relative to that of the landfast ice. The latter is mostly lacking such
areas of open water, rendering lateral melt and ocean-to-ice heat transfer from subsurface
ocean heat storage less effective (see also Petrich et al., 2012).
Table 5 summarizes the coastal-MASIE differences in break-up dates by grouping the sites
according to the role played by landfast ice. For several sites, the categorization of the fast ice
requires clarification.  The Chukchi Sea location is a non-coastal site and therefore clearly
beyond the extent of landfast ice (Figure 1). The St. Lawrence Island grid cells used here are
considered to be unaffected by land fast ice because of their location southeast of the island,
as described in Section 2.  The grid cells representing the Mestersvig region are located in the
coastal Greenland Sea, just outside of King Oscar Fjord. This region experiences dynamic ice
conditions with a comparatively short landfast ice season and a narrower landfast ice belt,
with ocean swell and ice pack interaction constraining extent and duration of the landfast ice
cover (Wadhams, 1981).  For this reason, Mestersvig is listed below the other sites affected by
landfast ice in Table 5.  With these caveats, it apparent from Table 5 that there is a general
tendency for later break-up (both the start and end dates) at locations affected by landfast ice.
The delay of the break-up ranges from about 5 to 40 days.  Exceptions are Pevek and Sabetta,
where local freshwater inflows from streams and snowmelt may contribute to earlier break-
ups relative to the broader MASIE regions – a hypothesis that should be tested in future
research. There is no clear signal of earlier or later coastal break-up at Mestersvig and St.
Lawrence Island, where fast ice is not a major contributor to the timing of break-up.  The
earlier local break-up at the Chukchi site is primarily a function of its location in the southern
portion of the Chukchi MASIE region.
Table 5.  Summary of landfast ice presence at each coastal site and timing of break-up at the
site relative to break-up in corresponding MASIE region (Figures 10 and 11).

| 650 | | Landfast ice? | Break-up start (vs. MASIE) | Break-up end (vs. MASIE) |
|---|---|---|---|---|
| 651 | Churchill | yes | later (~20 days) | similar |
| 652 | Clyde River | yes | later (~10 days) | later (~40 days) |
| 653 | Prudhoe Bay | yes | later (~15 days) | later (~15 days) |
| 654 | Utqiagvik | yes | latera (~10 days) | later (~15 days) |

| 655 | Tiksi | yes | later (~15 days) | similar |
| 656 | Pevek | yes | earlier (~5 days) | earlier (~5 days) |
| 657 | Sabetta | yes | similar | earlier (~15 days) |
| 658 | Mestersvig | (yes) | earlier (~20 days) | later (~15 days) |
| 659 | St. Lawrence I. | no | earlier (~5 days) | similar |
| 660 | Chukcbi Sea | no | earlier (~10 days) | earlier (~35 days) |


In the autumn, water in the shallow coastal areas cools more rapidly to the freezing point
because there is less stored heat below the surface. Coastal waters can also be fresher than
offshore waters because of terrestrial runoff that freshens the nearshore areas during the warm
season. Under such conditions both a higher freezing point and reduction of convective
overturning promote earlier freeze-up (Dmitrenko et al., 1999). As a result, the autumn freeze-
up often proceeds outward from the coast as well as shoreward from the main pack ice (Figure
12). However, onset of freeze-up – and depending on the geographic setting and offshore
ocean and atmosphere conditions potentially also end of freeze-up – do not correspond with
onset of landfast ice formation. In the Chukchi and Beaufort Sea, first appearance of landfast
ice may lag freeze onset by a couple of weeks to three months (Mahoney et al., 2014). In more
sheltered and less dynamic environments such as the Laptev Sea, inshore landfast ice typically
does not form for another couple of weeks after onset of freeze-up and generally takes more
than a month to extend further offshore (Selyyuzhenok et al., 2015). Hence, freeze-up
variability and trends reported in this study are seen as largely independent of landfast ice
processes.
Conversely, timing of freeze-up does impact the seasonal evolution of landfast ice. Mahoney
et al. (2007) discuss mean climatology of annual landfast ice from 1996-2004, including
analyses of the maximum, minimum and mean extents.  Notable for the results presented in
the present study is Mahoney et al.'s finding of a reduced presence of landfast ice in Beaufort-
Chukchi region, due to later formation and earlier breakup. In a follow-up study, Mahoney et
al. (2014) addressed the geographical variability of break-up and freeze-up, especially as it
relates to landfast ice.  Their results show that landfast ice in the central and western Beaufort
Sea forms earlier, breaks up later, occupies deeper water and extends further from shore than
that in the Chukchi Sea. These differences are partially due to the orientation of the coastline
relative to the prevailing easterly winds, which can more readily advect ice away from the
southwest-northeast oriented coastline of the Chukchi Sea. Hosekova et al. (2021) examined
landfast ice along the northern Alaska coast in the context of the buffering of the coastline
from wave activity.  They found that the wave attenuation by landfast ice was weaker in
autumn than in spring because of the lower ice thickness in autumn compared to spring.
However, the importance of waves for breakup is somewhat limited because it typically
requires large fetch with does not develop until later in the summer and fall, well past the end
of break-up season.
Yu et al. (2014) showed that landfast ice has large interannual variations, which imply large
variations in break-up and freeze-up. Superimposed on these variations were notable trends in
landfast ice during Yu et al's study period, 1976-2007.  More specifically, the duration of
landfast ice was found to have shortened in the Chukchi, East Siberian and Laptev Seas,
primarily as a result of a slower offshore expansion of landfast ice during the autumn and
early sinter since 1990. Our coastal sites in these sectors (Utqiagvik, Pevek and Tiksi) show
notable trends toward earlier break-up and later freeze-up, consistent with Yu et al.'s (2014)
trends in landfast ice.

Cooley et al. (2020) examined the sensitivity of landfast ice break-up at the community level
in the Canadian Arctic and western Greenland to temperature variations and trends based on
analysis of visible satellite imagery. Our analysis provides a longer reference period (40 years
vs. 19 years) and a broader geographical context for the work by Cooley and collaborators.
Cooley et al. (2020) also used the relationships between air temperature and landfast ice
break-up date, together with projected changes in air temperature from a set of eight CMIP5
global climate models, to project future changes in the breakup dates. Specifically, we note
that the trends projected for the remainder of the century in Cooley et al. (2020) are in many
instances less pronounced (in days/decade shift in breakup) than those identified here. For
example, for Clyde River Cooley et al. project a shift in breakup to an earlier date by 23 days
by the year 2099 as compared to changes of a similar magnitude but over a much shorter time
period examined here (Fig. 9 and 10). For Clyde River, the comparison between trends in the
local break-up timing compared to that for the broader region (Baffin Bay) also reveals that
the regional trends are much less pronounced than those at the local scale (Fig. 9 and 10).
Furthermore, the two westernmost communities examined by Cooley et al. (2020),
Tuktoyaktuk and Paulatuk (Eastern Beaufort Sea), were projected to see earlier landfast ice
break-up onset of 5 days and 11 days, respectively, by 2099. The data compiled here for
Prudhoe Bay and the Beaufort Sea indicate a substantially larger shift towards earlier dates by
more than 5 days *per decade* (Fig. 9 and 10).
One other study that addressed future changes of sea ice duration in the Pacific sector of the
Arctic is Wang et al.'s (2018) evaluation mid-21$^{st}$-century projections based on sea ice
concentrations simulated by seven CMIP5 global climate models. However, Wang et al.'s
evaluations were for the broader offshore areas of the East Siberian, Chukchi and Beaufort
Seas rather than for immediate coastal areas, as global climate models generally do not
include landfast ice. Pan-Arctic models that simulated landfast ice parameterized
thermodynamically without addressing its mobility had significant problems in forecasting
coastal ice thickness, especially during freeze-up in September and October (Johnson et al.,
2012). The projected increases in ice-free season length over the 2015-2044 period were
found were found to vary from about  20 days in the Bering Strait region to up to 60 days in
the offshore areas of the East Siberian, Chukchi and Beaufort Seas.  While these changes are
for offshore areas, they are larger than those projected for coastal areas by late century in the
study of Cooley et al. (2020).  .
**5.   Conclusion**
The primary objective of this study was to use the locally-based metrics to construct
indicators of break-up and freeze-up at near-coastal locations in which sea ice has high
stakeholder relevance.  A set of ten coastal locations distributed around the Arctic were
selected for this purpose. The sea ice indicators used here are based on local ice climatologies
informed by community ice use (Johnson and Eicken, 2016; Eicken et al., 2014) rather than
prescribed "universal" thresholds of ice concentration (e.g., 15%, 80%) used in other recent
studies of sea ice break-up and freeze-up.
The trends and interannual variations of the local indicators of break-up and freeze-up at the
ten nearshore are similar to the trends and variations of corresponding indicators for broader
offshore regions, but the site-specific indicators often differ from the regional indicators by
several days to several weeks. Relative to indicators for broader adjacent seas, the coastal
indicators show later break-up at sites known to have extensive landfast ice, whose break-up
typically lags retreat of the adjacent, thinner drifting ice. The coastal indicators also show an
earlier freeze-up at some sites in comparison with freeze-up for broader offshore regions,
likely tied to earlier freezing of shallow water regions and areas affected by freshwater input
from nearby streams and rivers. However, the trends towards earlier break-up and later freeze-
up are unmistakable over the post-1979 period at nearly all the coastal sites and their
corresponding regional seas.
The coastal indicators of the seasonal ice cycle for this study are based on Alaskan ice users.
However, ice uses and ice hazards in this region, as reflected in the definition of key seasonal
indicators, align with those of other coastal regions in the Arctic. Specifically, the
commonalities between coastal populations using the sea ice cover (both drifting and landfast)
as a platform for a range of activities, and to whom sea ice poses a hazard for boating and
marine vessel traffic, justify the approach taken in this study to extrapolate from the Alaskan
Arctic (with a range of ice conditions representative of the broader Arctic) to the pan-Arctic
scale.
The differences between the coastal and offshore regional indicators matter greatly to local
users whose harvesting of coastal resources and Indigenous culture are closely tied to the
timing of key events in the seasonal ice cycle (Huntington et al., 2021; Eicken et al., 2014).
These differences also matter from the perspective of maritime activities, where access to
coastal locations for destinational traffic is a key factor (Brigham, 2017). These offsets vary
considerably by region.   In light of these findings, we view locally as well as regionally
defined measures of sea-ice break-up and freeze-up as a key set of indicators linking pan-
Arctic or global indicators such as sea-ice extent or volume to local and regional uses of sea
ice, with the potential to inform community-scale adaptation and response.
**Acknowledgments**
This work was supported by the Climate Program Office of the National Oceanic and
Atmospheric Administration through Grant NA17OAR431060. Additional funding was
provided by the Interdisciplinary Research for Arctic Coastal Environments (InteRFACE)
project through the U.S. Department of Energy, Office of Science, Biological and
Environmental Research RGMA program.
**Data Availability**
The daily grids of passive-microwave-derived sea ice concentrations are available from the
National Snow and Ice Data Center as dataset NSIDC-0051, available at
https://nsidc.org/data/nsidc-0051.  Lists of the indicator dates for the coastal sites and the
MASIE regions are available from the author on request.
**Author contributions**
JEW served the principal investigator for the study, led the drafting of the manuscript, and
performed the factor analysis described in Section 3.  HE supervised the implementation of
the revised indicators for the coastal sites and the MASIE regions, and drafted parts of the
text.  KR performed the indicator calculations, produced Figures 1-11, and assisted in the
preparation of the manuscript.  MJ designed the original indicators, participated in the
modification of the indicators, and contributed to the revision of the manuscript.
**Competing interests**
The authors declare that they have no conflict of interest

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

**Supplementary material**

Table S1.  Dates (Julian day numbers) corresponding to the modal values (peaks) of the
distributions in Figure 4.  (Insufficient number of years met Bliss criteria in
Central Arctic).


| 966 967 | **Break-up Start** | | **Break-up end** | | **Freeze-up start** | | **Freeze-up end** | |
|---|---|---|---|---|---|---|---|---|
| 968 | J&E | Bliss | J&E | Bliss | J&E | Bliss | J&R | Bliss |
| 970 Beaufort Sea | 145 | 187 | 167 | 208 | 292 | 287 | 296 | 279 |
| 971 Chukchi Sea | 147 | 177 | 181 | 202 | 315 | 312 | 325 | 302 |
| 972 E. Sibarian Sea | 150 | 182 | 195 | 207 | 281 | 293 | 280 | 294 |
| 973 Laptev Sea | 140 | 192 | 188 | 207 | 280 | 271 | 285 | 279 |
| 974 Kara Sea | 145 | 193 | 190 | 209 | 304 | 299 | 307 | 296 |
| 975 Barents Sea | 146 | 164 | 152 | 186 | 315 | 297 | 328 | 302 |
| 976 Greenland Sea | 150 | 177 | 162 | 207 | 308 | 290 | 342 | 280 |
| 977 Baffin Bay | 121 | 152 | 149 | 186 | 331 | 311 | 346 | 324 |
| 978 Canadian Arctic | 147 | 208 | 190 | 207 | 279 | 274 | 298 | 275 |
| 979 Hudson Bay | 139 | 159 | 177 | 198 | 322 | 317 | 326 | 325 |
| 980 Central Arctic | 199 | | 200 | | 306 | | 310 | |
| 981 Bering Sea | 110 | 123 | 123 | 142 | 343 | 337 | 362 | 349 |









Table S2. Slopes (least-squares linear regression lines) of the MASIE regions in Figures 5-6
and 8-11. Also shown are the explained variances ($r^2$ values of the trend lines and their levels
of statistical significance.

| Region | Indicator Group | Indicator | Slope (days yr$^{-1}$) | $r^2$ | significance level |
|---|---|---|---|---|---|
| | | | | | |
| Baffin Bay | Bliss | Day of Advance | 0.4 | 0.57 | < 0.01** |
| Baffin Bay | Bliss | Day of Closing | 0.4 | 0.52 | < 0.01** |
| Baffin Bay | Bliss | Day of Opening | -0.5 | -0.74 | < 0.01** |
| Baffin Bay | Bliss | Day of Retreat | -0.7 | -0.77 | < 0.01** |
| Baffin Bay | J&E | Break-up End | -0.2 | -0.44 | < 0.01** |
| Baffin Bay | J&E | Break-up Start | -0.1 | -0.07 | 0.67 |
| Baffin Bay | J&E | Freeze-up End | 0.4 | 0.57 | < 0.01** |
| Baffin Bay | J&E | Freeze-up Start | 0.5 | 0.71 | < 0.01** |
| Barents Sea | Bliss | Day of Advance | 1.3 | 0.7 | < 0.01** |
| Barents Sea | Bliss | Day of Closing | 1.3 | 0.7 | < 0.01** |
| Barents Sea | Bliss | Day of Opening | -1.1 | -0.72 | < 0.01** |
| Barents Sea | Bliss | Day of Retreat | -1.2 | -0.79 | < 0.01** |
| Barents Sea | J&E | Break-up End | -1.0 | -0.72 | < 0.01** |
| Barents Sea | J&E | Break-up Start | -0.4 | -0.38 | 0.02* |
| Barents Sea | J&E | Freeze-up End | 1.0 | 0.72 | < 0.01** |
| Barents Sea | J&E | Freeze-up Start | 1.0 | 0.8 | < 0.01** |
| Beaufort Sea | Bliss | Day of Advance | 0.8 | 0.61 | < 0.01** |
| Beaufort Sea | Bliss | Day of Closing | 0.9 | 0.63 | < 0.01** |
| Beaufort Sea | Bliss | Day of Opening | -0.7 | -0.51 | < 0.01** |
| Beaufort Sea | Bliss | Day of Retreat | -1.0 | -0.56 | < 0.01** |
| Beaufort Sea | J&E | Break-up End | -0.7 | -0.48 | < 0.01** |
| Beaufort Sea | J&E | Break-up Start | -0.6 | -0.51 | < 0.01** |

| | | | | | |
|---|---|---|---|---|---|
| Beaufort Sea | J&E | Freeze-up End | 0.7 | 0.68 | < 0.01** |
| Beaufort Sea | J&E | Freeze-up Start | 0.7 | 0.65 | < 0.01** |
| Bering Sea | Bliss | Day of Advance | 0.4 | 0.43 | < 0.01** |
| Bering Sea | Bliss | Day of Closing | 0.4 | 0.36 | 0.02* |
| Bering Sea | Bliss | Day of Opening | -0.2 | -0.28 | 0.09 |
| Bering Sea | Bliss | Day of Retreat | -0.3 | -0.37 | 0.02* |
| Bering Sea | J&E | Break-up End | -0.0 | -0.01 | 0.98 |
| Bering Sea | J&E | Break-up Start | 0.0 | 0.05 | 0.77 |
| Bering Sea | J&E | Freeze-up End | 0.3 | 0.33 | 0.04* |
| Bering Sea | J&E | Freeze-up Start | 0.5 | 0.65 | < 0.01** |
| Canadian Arch. | Bliss | Day of Advance | 0.5 | 0.63 | < 0.01** |
| Canadian Arch. | Bliss | Day of Closing | 0.6 | 0.56 | < 0.01** |
| Canadian Arch. | Bliss | Day of Opening | -0.3 | -0.57 | < 0.01** |
| Canadian Arch. | Bliss | Day of Retreat | -0.9 | -0.7 | < 0.01** |
| Canadian Arch. | J&E | Break-up End | -0.4 | -0.62 | < 0.01** |
| Canadian Arch. | J&E | Break-up Start | -0.4 | -0.5 | < 0.01** |
| Canadian Arch. | J&E | Freeze-up End | 0.3 | 0.58 | < 0.01** |
| Canadian Arch. | J&E | Freeze-up Start | 0.2 | 0.51 | < 0.01** |
| Central Arctic | Bliss | Day of Closing | 0.7 | 0.33 | 0.04* |
| Central Arctic | Bliss | Day of Opening | -0.5 | -0.17 | 0.31 |
| Central Arctic | J&E | Break-up End | -1.0 | -0.36 | 0.03* |
| Central Arctic | J&E | Break-up Start | -0.9 | -0.31 | 0.06 |
| Central Arctic | J&E | Freeze-up End | 0.1 | 0.03 | 0.88 |
| Central Arctic | J&E | Freeze-up Start | 0.6 | 0.18 | 0.31 |
| Chukchi Sea | Bliss | Day of Advance | 1.0 | 0.75 | < 0.01** |
| Chukchi Sea | Bliss | Day of Closing | 1.1 | 0.73 | < 0.01** |

| Chukchi Sea | Bliss | Day of Opening | -0.7 | -0.71 | < 0.01** |
|---|---|---|---|---|---|
| Chukchi Sea | Bliss | Day of Retreat | -0.7 | -0.66 | < 0.01** |
| Chukchi Sea | J&E | Break-up End | -0.6 | -0.65 | < 0.01** |
| Chukchi Sea | J&E | Break-up Start | -0.5 | -0.46 | < 0.01** |
| Chukchi Sea | J&E | Freeze-up End | 0.8 | 0.69 | < 0.01** |
| Chukchi Sea | J&E | Freeze-up Start | 1.0 | 0.79 | < 0.01** |
| E. Siberian Sea | Bliss | Day of Advance | 0.8 | 0.74 | < 0.01** |
| E. Siberian Sea | Bliss | Day of Closing | 1.1 | 0.78 | < 0.01** |
| E. Siberian Sea | Bliss | Day of Opening | -0.7 | -0.51 | < 0.01** |
| E. Siberian Sea | Bliss | Day of Retreat | -0.8 | -0.6 | < 0.01** |
| E. Siberian Sea | J&E | Break-up End | -0.5 | -0.45 | < 0.01** |
| E. Siberian Sea | J&E | Break-up Start | -0.7 | -0.46 | < 0.01** |
| E. Siberian Sea | J&E | Freeze-up End | 0.6 | 0.76 | < 0.01** |
| E. Siberian Sea | J&E | Freeze-up Start | 0.7 | 0.77 | < 0.01** |
| Greenland Sea | Bliss | Day of Advance | 0.9 | 0.62 | < 0.01** |
| Greenland Sea | Bliss | Day of Closing | 0.5 | 0.45 | < 0.01** |
| Greenland Sea | Bliss | Day of Opening | -0.4 | -0.38 | 0.02* |
| Greenland Sea | Bliss | Day of Retreat | -0.6 | -0.5 | < 0.01** |
| Greenland Sea | J&E | Break-up End | -0.3 | -0.32 | 0.05* |
| Greenland Sea | J&E | Break-up Start | -0.0 | -0.04 | 0.79 |
| Greenland Sea | J&E | Freeze-up End | 0.4 | 0.38 | 0.02* |
| Greenland Sea | J&E | Freeze-up Start | 0.7 | 0.63 | < 0.01** |

| | | | | | |
|---|---|---|---|---|---|
| Hudson Bay | Bliss | Day of Advance | 0.5 | 0.64 | < 0.01** |
| Hudson Bay | Bliss | Day of Closing | 0.4 | 0.57 | < 0.01** |
| Hudson Bay | Bliss | Day of Opening | -0.5 | -0.67 | < 0.01** |
| Hudson Bay | Bliss | Day of Retreat | -0.7 | -0.74 | < 0.01** |
| Hudson Bay | J&E | Break-up End | -0.4 | -0.65 | < 0.01** |
| Hudson Bay | J&E | Break-up Start | -0.1 | -0.06 | 0.72 |
| Hudson Bay | J&E | Freeze-up End | 0.4 | 0.55 | < 0.01** |
| Hudson Bay | J&E | Freeze-up Start | 0.6 | 0.73 | < 0.01** |
| Kara Sea | Bliss | Day of Advance | 0.7 | 0.63 | < 0.01** |
| Kara Sea | Bliss | Day of Closing | 0.9 | 0.66 | < 0.01** |
| Kara Sea | Bliss | Day of Opening | -1.0 | -0.75 | < 0.01** |
| Kara Sea | Bliss | Day of Retreat | -1.1 | -0.76 | < 0.01** |
| Kara Sea | J&E | Break-up End | -0.9 | -0.7 | < 0.01** |
| Kara Sea | J&E | Break-up Start | -0.3 | -0.22 | 0.18 |
| Kara Sea | J&E | Freeze-up End | 0.8 | 0.62 | < 0.01** |
| Kara Sea | J&E | Freeze-up Start | 0.7 | 0.64 | < 0.01** |
| Laptev Sea | Bliss | Day of Advance | 0.6 | 0.65 | < 0.01** |
| Laptev Sea | Bliss | Day of Closing | 0.7 | 0.64 | < 0.01** |
| Laptev Sea | Bliss | Day of Opening | -0.6 | -0.55 | < 0.01** |
| Laptev Sea | Bliss | Day of Retreat | -0.7 | -0.58 | < 0.01** |
| Laptev Sea | J&E | Break-up End | -0.6 | -0.52 | < 0.01** |
| Laptev Sea | J&E | Break-up Start | -0.7 | -0.48 | < 0.01** |
| Laptev Sea | J&E | Freeze-up End | 0.4 | 0.68 | < 0.01** |
| Laptev Sea | J&E | Freeze-up Start | 0.4 | 0.64 | < 0.01** |
| | | | | | |



Table S3.  Same as Table S2, but for the local indicators.  Slopes (linear regression lines)
correspond to Figures 8-11. Also shown are the explained variances ($r^2$ values of the trend
lines and their levels of statistical significance.

| Location | Indicator Group | Indicator | Slope (days yr$^{-1}$) | $r^2$ | Significance level |
|----------|-----------------|-----------|------------------------|-------|--------------------|
|  |  |  |  |  |  |
| Churchill | Bliss | Day of Advance | 0.3 | 0.52 | < 0.01** |
| Churchill | Bliss | Day of Closing | 0.4 | 0.51 | < 0.01** |
| Churchill | Bliss | Day of Opening | −0.8 | −0.59 | < 0.01** |
| Churchill | Bliss | Day of Retreat | −1.0 | −0.67 | < 0.01** |
| Churchill | J&E | Break-up End | −0.7 | −0.54 | < 0.01** |
| Churchill | J&E | Break-up Start | −0.5 | −0.3 | 0.07 |
| Churchill | J&E | Freeze-up End | 0.4 | 0.49 | < 0.01** |
| Churchill | J&E | Freeze-up Start | 0.7 | 0.53 | < 0.01** |
| Clyde River | Bliss | Day of Advance | 0.3 | 0.46 | < 0.01** |
| Clyde River | Bliss | Day of Closing | 0.3 | 0.45 | < 0.01** |
| Clyde River | Bliss | Day of Opening | −0.6 | −0.47 | < 0.01** |
| Clyde River | Bliss | Day of Retreat | −0.5 | −0.42 | < 0.01** |
| Clyde River | J&E | Break-up End | −0.6 | −0.5 | < 0.01** |
| Clyde River | J&E | Break-up Start | −0.5 | −0.22 | 0.18 |
| Clyde River | J&E | Freeze-up End | 0.3 | 0.45 | < 0.01** |
| Clyde River | J&E | Freeze-up Start | 0.3 | 0.43 | < 0.01** |
| Mestersvig | Bliss | Day of Advance | 0.6 | 0.36 | 0.05* |
| Mestersvig | Bliss | Day of Closing | 0.9 | 0.52 | < 0.01** |
| Mestersvig | Bliss | Day of Opening | −0.7 | −0.36 | 0.02* |
| Mestersvig | Bliss | Day of Retreat | −0.6 | −0.37 | 0.04* |
| Mestersvig | J&E | Break-up End | −0.2 | −0.2 | 0.26 |
| Mestersvig | J&E | Break-up Start | 0.1 | 0.04 | 0.83 |

| | | | | | |
|---|---|---|---|---|---|
| Mestersvig | J&E | Freeze-up End | 0.6 | 0.5 | < 0.01** |
| Mestersvig | J&E | Freeze-up Start | 0.5 | 0.42 | 0.02* |
| Pevek | Bliss | Day of Advance | 1.1 | 0.72 | < 0.01** |
| Pevek | Bliss | Day of Closing | 1.1 | 0.77 | < 0.01** |
| Pevek | Bliss | Day of Opening | -0.9 | -0.4 | 0.01* |
| Pevek | Bliss | Day of Retreat | -1.0 | -0.46 | < 0.01** |
| Pevek | J&E | Break-up End | -0.7 | -0.33 | 0.05 |
| Pevek | J&E | Break-up Start | -1.1 | -0.37 | 0.03* |
| Pevek | J&E | Freeze-up End | 0.8 | 0.76 | < 0.01** |
| Pevek | J&E | Freeze-up Start | 0.9 | 0.73 | < 0.01** |
| Prudhoe Bay | Bliss | Day of Advance | 0.8 | 0.52 | < 0.01** |
| Prudhoe Bay | Bliss | Day of Closing | 0.8 | 0.65 | < 0.01** |
| Prudhoe Bay | Bliss | Day of Opening | -1.0 | -0.56 | < 0.01** |
| Prudhoe Bay | Bliss | Day of Retreat | -0.9 | -0.51 | < 0.01** |
| Prudhoe Bay | J&E | Break-up End | -0.8 | -0.54 | < 0.01** |
| Prudhoe Bay | J&E | Break-up Start | -0.5 | -0.27 | 0.1 |
| Prudhoe Bay | J&E | Freeze-up End | 0.8 | 0.6 | < 0.01** |
| Prudhoe Bay | J&E | Freeze-up Start | 0.7 | 0.59 | < 0.01** |
| Sabetta | Bliss | Day of Advance | 0.4 | 0.55 | < 0.01** |
| Sabetta | Bliss | Day of Closing | 0.4 | 0.47 | < 0.01** |
| Sabetta | Bliss | Day of Opening | -0.9 | -0.59 | < 0.01** |
| Sabetta | Bliss | Day of Retreat | -1.0 | -0.78 | < 0.01** |
| Sabetta | J&E | Break-up End | -0.8 | -0.56 | < 0.01** |
| Sabetta | J&E | Break-up Start | -0.9 | -0.42 | < 0.01** |
| Sabetta | J&E | Freeze-up End | 0.4 | 0.41 | < 0.01** |
| Sabetta | J&E | Freeze-up Start | 0.4 | 0.56 | < 0.01** |

| South Chukchi Sea | Bliss | Day of Advance | 0.9 | 0.63 | < 0.01** |
|---|---|---|---|---|---|
| South Chukchi Sea | Bliss | Day of Closing | 0.7 | 0.58 | < 0.01** |
| South Chukchi Sea | Bliss | Day of Opening | −0.6 | −0.51 | < 0.01** |
| South Chukchi Sea | Bliss | Day of Retreat | −0.7 | −0.56 | < 0.01** |
| South Chukchi Sea | J&E | Break-up End | −0.6 | −0.52 | < 0.01** |
| South Chukchi Sea | J&E | Break-up Start | −0.6 | −0.39 | 0.02* |
| South Chukchi Sea | J&E | Freeze-up End | 0.7 | 0.57 | < 0.01** |
| South Chukchi Sea | J&E | Freeze-up Start | 0.8 | 0.63 | < 0.01** |
| St. Lawrence Island | Bliss | Day of Advance | 0.6 | 0.33 | 0.05* |
| St. Lawrence Island | Bliss | Day of Closing | 0.3 | 0.2 | 0.24 |
| St. Lawrence Island | Bliss | Day of Opening | −0.1 | −0.16 | 0.35 |
| St. Lawrence Island | Bliss | Day of Retreat | −0.3 | −0.28 | 0.09 |
| St. Lawrence Island | J&E | Break-up End | −0.1 | −0.11 | 0.49 |
| St. Lawrence Island | J&E | Break-up Start | −0.0 | −0.02 | 0.92 |
| St. Lawrence Island | J&E | Freeze-up End | 0.4 | 0.25 | 0.13 |
| St. Lawrence Island | J&E | Freeze-up Start | 0.5 | 0.33 | 0.04* |
| Tiksi | Bliss | Day of Advance | 0.2 | 0.36 | 0.02* |
| Tiksi | Bliss | Day of Closing | 0.2 | 0.41 | 0.01* |

| Tiksi | Bliss | Day of Opening | −0.4 | −0.54 | < 0.01** |
|---|---|---|---|---|---|
| Tiksi | Bliss | Day of Retreat | −0.6 | −0.54 | < 0.01** |
| Tiksi | J&E | Break-up End | −0.3 | −0.53 | < 0.01** |
| Tiksi | J&E | Break-up Start | −0.3 | −0.34 | 0.03* |
| Tiksi | J&E | Freeze-up End | 0.3 | 0.45 | < 0.01** |
| Tiksi | J&E | Freeze-up Start | 0.2 | 0.45 | < 0.01** |
| Utqiaƒ°vik | Bliss | Day of Advance | 1.1 | 0.6 | < 0.01** |
| Utqiaƒ°vik | Bliss | Day of Closing | 1.1 | 0.67 | < 0.01** |
| Utqiaƒ°vik | Bliss | Day of Opening | −1.2 | −0.52 | < 0.01** |
| Utqiaƒ°vik | Bliss | Day of Retreat | −1.2 | −0.71 | < 0.01** |
| Utqiaƒ°vik | J&E | Break-up End | −0.7 | −0.52 | < 0.01** |
| Utqiaƒ°vik | J&E | Break-up Start | −0.7 | −0.27 | 0.11 |
| Utqiaƒ°vik | J&E | Freeze-up End | 0.8 | 0.66 | < 0.01** |
| Utqiaƒ°vik | J&E | Freeze-up Start | 0.9 | 0.62 | < 0.01** |
| | | | | | |




