# Peer review of "Sea ice break-up and freeze-up indicators for users"

_The Cryosphere, 2022_

## Author Comment (AC2)

**RC2**: ['Comment on tc-2022-21'](), Anonymous Referee #2, 11 Mar 2022

Reviewer comments are in regular font; *author responses in italics*

Review of Walsh et al., "Sea ice break-up and freeze-up indicators for users of the Arctic coastal environment," submitted for publication in The Cryosphere.

This manuscript describes a nice study focused on the difference in ice break-up and freeze-up dates between near-coastal locations vs larger scale regional averages. The paper is generally well-written and the figures are clear. The results are convincing and will have an impact on how such analyses are used for coastal sea ice science. I do have some questions and comments, so I recommend for publication with major revisions.

Lines 63-64: As noted below, Bliss does not use this type of definition. So there are a variety of methods.

*We will modify the text to say "Sea ice concentration thresholds have been used in various studies to determine the dates of sea ice opening, retreat, advance and closing. For example, Bliss et al., (2019) define dates of opening and retreat as, respectively, the last days on which the sea ice concentration drops below 80% and 15% before the summer minimum. Corresponding metrics are used by Bliss et al. for the dates of advance and closing."*

Lines 77-78: The 25 km PMW products provide the longest time record. However, we have at least 20 years of higher resolution AMSR data as well. Do you think your analysis would differ if you used that? I suggest you acknowledge that higher resolution PMW data are available and that they might be uniquely useful for examining coastal science.

*The higher-resolution AMSR data would resolve the near-shore sea ice distribution that is a focus of this study. We will add the following after lines 77-78 (original line numbers): "While higher-resolution permitting finer resolution of coastal sea ice is available from sensors such as AMSR (Advanced Microwave Radiometric Sounder), the record lengths are sufficiently shorter (about 20 years for AMSR) that our trend analysis would be severely limited by a reliance on such products".*

Line 94: You've got a typo here. NSIDC-0051 is the NASA Team data set. You want to write NSIDC-G02202, the CDR data set. And also note that there is a Version 4 available: was this not available when you started this work?

*We did use the data from NSIDC-0051. We will delete the reference to the NOAA/NSIDC Climate Data Record. The documentation for this dataset ([https://nsidc.org/data/nsidc-0051](https://nsidc.org/data/nsidc-0051))*

*points to the use of the NASA Team algorithm. The analysis described here was initiated prior to the availability of Version 4 of NSIDC-G02202.*

Line 107: "Examples include…" I suggest you clearly write out all modifications from the previous algorithm, so your methods can be potentially reproduced by others.

*This comment is similar to one from Reviewer 1. In response, we will add a paragraph (following the previous Line 112) summarizing the changes in each indicator relative to the criteria used by Johnson and Eicken (2016). In brief, the following modifications were incorporated into the algorithms:*

*Break-up start:*
*- minimum sic threshold created at 15% (vs. last day exceeding Jan-Feb mean minus 2σ)*
*- undefined if average summer sic > 40% (vs. no such criterion in J&E)*
*- undefined if subsequent breakup end date not defined*

*Break-up end:*
*- first time sic below threshold for 2 weeks instead of last day below threshold*
  *(vs. last exceeding larger of Aug-Sep mean or 15%)*
*- minimum threshold 50% instead of 15%*
*- undefined if break-up start not defined*

*Freeze-up start:*
*- first day on which sic exceeds Aug-Sep average by 1σ (as in J&E)*
*- undefined if mean summer sic > 25%*
*- undefined if subsequent freeze-up end not defined*

*Freeze-up end:*
*- first time sic above threshold for following 2 weeks instead of first day above threshold*
  *(threshold is Jan-Feb average minus 10%, as in J&E)*
*- thresholds imposed: Minimum (15%) and maximum (50%)*
*- undefined if sic always exceeds threshold*

Lines 110-111: I don't think v4 of the CDR has missing days.

*See response to comment on Line 94. We did not use the CDR.*

Line 111: "a spatial and then temporal smoothing" Please provide all details.

*The following will be added in the revised manuscript: "The daily sea ice concentration values were spatially smoothed using a generic mean filter with a square footprint of 3 x 3 grid cells. This data was then temporally smoothed three times in sequence using a Hann window"*

Table 1: There are a large number of seemingly arbitrary numbers here. Why 25%, or 40%, or 50%? Why two standard deviations, or one standard deviation? Why minus 10%? First, why did you pick these, and second, what if you vary these numbers?

*The 10% threshold is based on prior work (Johnson and Eicken, 2016) in which sensitivities were explored. The 25%, 40% and 50% thresholds in Table 1 were arrived at by testing various values and selecting values that maximized the number of years with break-up and freeze-up dates defined. The selected values were those that generally maximized the number of such years across the various coastal locations and MASIE regions.*

\* You are providing the lat/lon of these coastal communities/stations. Why? They are not used anywhere. Instead, perhaps you can provide the exact grid cell numbers of the 3 cells for each location.

*In the interest of a general readership (one that does not work specifically with the NSIDC passive microwave products), we believe that the latitude and longitude will be more useful to readers than the grid cell coordinates. The latter have the disadvantage that they will vary with the sea ice dataset and its resolution, while the latitude and longitude are site-specific and fixed.*

\* Mestersvig seems to be decommissioned, as far as I can tell. Why did you choose this?

*While the permanent military function at Mestersvig has been discontinued, the site has been used as recently as 2015 for military exercises to test the responsiveness of Denmark's military and the cold-weather functionality of its equipment. Since the airport (with its 1800 meter airstrip) and other infrastructure, further use of this site is possible, if not probable.*

Figure 1:

\* Could you rotate all panels so that north is upward, as is usual?

*In a revised version of this figure, we have rotated the panels for consistent north-south orientation and have added a northward-pointing "N" arrow.*

* "Bering Strait coast" is an odd name for this location. Really it is "NE Chukchi Sea," right? Also, these 3 grid cells are farther away from the coast than the other locations. Why?

*This location is actually in the southern Chukchi Sea, as will be apparent in our revised Figure 1, which (per the suggestion of Reviewer 1) shows all the locations listed in Table 2. We have changed the name of this site in the Table to simply "Chukchi Sea". This location is indeed farther from the coast than the other sites, and the location was intentionally selected to be farther offshore in order to provide a non-coastal counter-example to the other sites, all of which are adjacent to a coast.*

* I suggest adding distance or lat/lon markers along the horizontal and vertical axes of each panel.

*As noted above (2nd preceding bullet point), we have reoriented all panels in Figure 1 so that north is consistently at the top of each panel. We have also added a distance scale indicator for the entire figure*

More on Figure 1: It took me a few minutes to understand the basic strategy here, so I suggest that you make this clearer. You show these coastal locations with red dots, but really they are never used. However, this distracted me and I thought you were going to compare coastal station information with these three grid cells. No, in fact you are using these 3 grid cells as proxies for coastal conditions. I suggest you say this explicitly. And then I am wondering:

*We will expand the caption to state explicitly that the red dots (black dots in revised version) represent the actual locations of coastal communities, while the offshore squares represent the grid cells of the passive-microwave-derived sea ice concentrations used in computing the break-up and freeze-up metrics, expecting them to be reasonable estimates for the sea-ice indicators at each coastal location.*

1) Why not choose SIC cells right at the coast? Yes, there could be coastal contamination, but I think the CDR has eliminated or much reduced this. At least you should check this.

*In our examination of the coastal regions in NSIDC-0051, we found that there is indeed still some contamination present in grid cells adjacent to the coastline. For this reason, we chose grid cells that were as close as possible to the coastline without being at the coast (i.e., we required that there be a one-cell offshore displacement of any selected grid cell).*

2) Why not validate these CDR SIC values with coastal information when available? EG Utqiaġvik has a sea ice radar, doesn't it?

*There is a sea ice radar at Utqiaġvik, but its range is less than 25 km so it does not "see" beyond the grid cell immediately adjacent to the coast. The northern Alaska coast is an example of a region in which coastal contamination led to non-zero ice concentrations at coastal grid cells when sea ice was not present (e.g., mid-September of years in the final decade of the dataset).*

3) What if you chose a trio of CDR SIC cells that were at another location near the coast within a MASIE region? Do you think you would get a different result? I suggest that you take one or two MASIE regions and test this, ie take 3 different near-coastal locations within a MASIE region and run your analysis. How different are your results?

*In response to this comment, we actually experimented with the grid cell selections at Utqiaġvik and Sabetta. The trends and interannual variations of the break-up start and freeze-up end showed no notable changes when the three selected grid cells were displaced offshore or alongshore by one grid cell.*

4) You interpret many results in the context of landfast ice. I think you need to show that the CDR SIC cells you have chosen (because they are offshore) are actually in the landfast ice zone at each location. EG you could add a typical landfast ice contour (when it's there) to each panel of Figure 1. Or better, some kind of interannual min/max measure of landfast ice extent relative to the cells.

*This concern was also raised by Reviewer 1. In response, we are augmenting this section with additional discussion (placed in a new Discussion section, as suggested by Reviewer 1) and a new figure showing the maximum spatial extent of landfast ice during June (the middle of the break-up period). This map is based on the charts of the U.S. National Ice Center for the 35 years ending 2007. We have superimposed on this map the locations of our coastal sites, thereby providing a basis for our assessment of a role of landfast ice in delaying the break-up.*

Line 163: Typo: "Figure 1" -> "Figure 2"

*Will be corrected in revision.*

Lines 165-167: Why did you include MASIE regions that are not near your 10 locations, eg "Central Arctic?"

*The original text was intended to list the entire set of Arctic and subarctic regions in Figure 2. In the revision, we will omit from these lines the regions that we do not use in this study (Regions 6, 9 and 11).*

Line 193: Typo: "Table 2" -> "Table 1"

*Will be corrected in revision.*

Figure 3: It might be useful to remind the reader in this caption that break-up start only exists when there's break-up end, and same for freeze-up, thus only two panels are needed in this figure.

*This reminder will be added to the caption of Figure 3.*

Lines 213-216: This is interesting; I think Bliss vs J&E is in some ways analogous to NASA Team vs NASA Bootstrap. IE Team uses fixed tie points, while Bootstrap uses "dynamic" ie time/space varying tie points. There is value in each method (although generally the latter seems to be more accurate).

*We will add a statement pointing to this analogy between the algorithms and the J&E vs. Bliss differences.*

Figure 4: I had to look up "violin plots" but there is a Wikipedia page so ok. Still, it might be nice to write one sentence explaining what this is. Also:

*We will restructure the opening of the paragraph (originally beginning on Line 232) to explain what a violin plot actually is.*

* What are the black strips within each histogram? Are they gaps where there's no data for that day of year?

*The thin black lines represent the observations themselves, i.e., the indicator dates for each year. The black strips are clusters of lines representing groups of similar values in the distribution, i.e., in this case they represent a narrow range of dates in which multiple years had their break-up or freeze-up dates. We will add this information near the start of the paragraph introducing the violin plots.*

* Perhaps you want to mark the mode or mean on each histogram, and write it as text?

*We are adding a new table (Table 3) listing the mode (peak of distribution) of the dates in the violin plots  For each metric and location, the table includes the corresponding values from J&E and Bliss et al.*

Figure 5: I suggest writing in text the two slopes in each panel within each panel (or you could create a separate table). And this goes for all subsequent similar figures with trend lines. Perhaps one big new table with all slopes for all such figures together?

*Since legible font would clutter the 12-panel figures, we will list the slopes a new table (Table 4) for Figures 5-6 and a similar table (Table 5) for Figures 8-11.*

Line 377: Typo: "…and negative…" ie add "and"

*"and" will be inserted in revision.*

Line 387: Typo: "Figure 12" -> "Figure 13"

*Will be corrected to "Figure 13" in revision.*

Line 411: Here is a main result: The variance at three grid cells within a large region is higher than for the regional mean. This is useful in the context of coastal science, and it is very reasonable i.e., intuitive. I might propose that it is too reasonable, i.e., isn't this expected that variance increases when you focus on a small subregion? Actually, this could be true in some cases and the opposite might be true in others (eg if you picked three grid cells in a "boring" place with low variance). What if you took a MASIE region and made a map of some factor analysis parameter that would illustrate this? Would it show higher variance near the coast and lower variance toward the perennial ice pack? This seems important for you in order to really show that your result is not so obvious.

*We had some difficulty in interpreting this comment, but we tried to address it by performing a factor analysis on the ice concentrations within a MASIE region (Figure 1's Region 2, the Chukchi Sea).  The strongest factor loadings were found in the south, where more years experience freeze-up and break-up than in the north. As the reviewer notes, this is not surprising because the variance is greater in the south than in the north.  We will add a statement in the text*

*after the statement beginning Line 411 (original submission), pointing out that the difference in variance indeed contributes to the difference in explained for the local and regional indicators. However, we do not find it so obvious that the percentage of variance explained by the first factor should be greater for the more localized metrics.*

---

## Author Response (AR1)

Reviewer comments are in regular font; *our responses in italics*

**Reviewer 1**

**RC1**: 'Comment on tc-2022-21', Anonymous Referee #1, 09 Mar 2022

**To the Authors,**

This is an interesting and relevant study that recognizes how coastal zones possess unique sea ice regimes, and works to develop more suitable indicators to identify freeze up and break up. Your methods are promising for future studies to adopt in coastal zones experiencing seasonal ice cover, and the broad spatial and temporal coverage of this study makes it applicable to a broad audience of scientists focusing on sea ice from the Laptev Sea to Baffin Bay. However, this paper has significant problems that I recommend are addressed before publication. These problems are explained in detail in my line-by-line feedback (below), but the decision-making that went in to cell placement in the MASIE regions are not clear, which has a ripple effect in how the results are interpreted, since the freeze-up and break-up that occurs within these cells (covering no greater than 75 square kilometers in surface area) are interpreted in the broader geographical context of the MASIE regions they fall under. Until the methodology is better explained for why the cell locations were selected, the subsequent interpretation of results remains weak. It also inhibits the replicability of these methods in future studies. In the results, correlations are assumed, justified with qualitative descriptions of the coastal sea ice regimes. For example, the later break up dates are attributed to the lag in landfast ice break up compared to drift ice. However, there is no attempt to validate (through existing datasets or literature, see feedback on line 457) that the later break ups in these cells are indeed landfast ice, and not some other coastal process unique to the region of study. Correlations between freeze up and break up trends and coastal geography needs to have been determined by yourself, or strongly supported by citing previous studies. Results interpretation is also an issue, as this generally occurs in the Results section without citations, and a Discussion section is not included. The significance of your findings is not impactful without better referencing of the literature, and neglecting to do so inhibits the contribution this makes toward expanding the body of knowledge about the cryosphere, which is the stated purpose of TC as a journal. These problems must be corrected with additional writing and reorganization of the manuscript, as well as potentially requiring additional data analysis for validation of results interpretation if it is not supportable by previous studies. Therefore, I recommend this manuscript be accepted pending major revisions.

*The reviewer lists three major concerns that need to be addressed: (1) the basis for the selection of locations for cell placement, (2) the validation of the claims about the role of landfast ice and (3) the need for additional references to the published literature in the Results section. The revised manuscript responds to each of these concerns, as described below in the point-by-point responses. In particular, we have added clarification and justification for the selection of the grid cells offshore of each local site; we have prepared a new section on landfast ice (including a figure on its spatial distribution and the addition of landfast ice masks to the panels in Figure 1) to support our claims about the role of landfast ice in the indicator offset; and we have added a new Discussion section (Section 4) in order to provide context for our findings. The revision*

*contains 22 additional references, more than doubling the number in the original submission, to help provide this context.*

**Line by Line feedback**

Line 53: A brief definition of how we define "offshore" would be useful for the uninitiated. Is this a certain distance from the coastline? A bathymetric contour?

*"Offshore" was used here to refer to the nearshore marine environment because it is often used by the stakeholders in coastal communities. The wording has been changed to "nearshore marine" in the revised manuscript to make it clearer to TC readers.*

Line 107: I could be misunderstanding this paragraph, but are the authors saying that these definitions are revised to be more appropriate to offshore environments, which is the focus of this study? If so, why are the authors focused on revising criteria to broaden the applicability to "non-coastal" areas, which are not the focus of this study? Or is this a typo of 'near-coastal', which is a term used in this paper? In any case, what's clear is the authors are revising these metrics to consider break up and freeze up in an environment that was previously unaccounted for. In which case I would suggest an explanation of what the criteria used to be, how the criteria has been changed in this study, and how this change is better suited for the environment the new criteria is being applied to. This would probably work best as additional columns in table 1, where information on the revised criteria is already provided.

*There was indeed a typo, as the text was referring to coastal areas. The typo has been corrected in the revision. In response to the second part of the comment, we have added a new paragraph and table (Table 3) summarizing the changes in each indicator relative to the criteria used by Johnson and Eicken (2016). In brief, the following modifications of the J&E criteria were incorporated into the algorithms (sic = sea ice concentration; σ = standard deviation):*

*Break-up start:*
*- minimum sic threshold created at 15% (vs. J&E's last day exceeding Jan-Feb mean minus 2σ)*
*- undefined if average summer sic > 40% (vs. no such criterion in J&E)*
*- undefined if subsequent breakup end date not defined*

*Break-up end:*
*- first time sic is below threshold (larger of Aug-Sep mean or 15% for 2 weeks)*
  *(vs. first day below threshold in J&E)*
*- minimum threshold 50% instead of 15%*
*- undefined if break-up start not defined*

*Freeze-up start:*
*- first day on which sic exceeds Aug-Sep average by 1σ (as in J&E)*

*- undefined if mean summer sic > 25%*
*- undefined if subsequent freeze-up end not defined*

*Freeze-up end:*
*- first time sic above threshold for following 2 weeks instead of first day above threshold*
   *(threshold is Jan-Feb average minus 10%, as in J&E)*
*- thresholds imposed: Minimum (15%) and maximum (50%)*
*- undefined if sic always exceeds threshold*

Line 113: This paper is written with a clear objective in mind and would benefit from a consistent articulation of it to be used throughout the paper. There are three areas where an objective is articulated, but the wording varies. Line 88-90 states the objective of this paper is to examine the timing of freeze/break up as key constraints for human/ecosystem activity. Line 113 says a key objective of the work is to compare various dates at nearshore locations with corresponding metrics for broader (non-coastal?) areas of the AO and subarctic seas. Line 449 says the primary objective is to use local-based indicators to construct freeze/break up indicators at near-coastal locations relevant to stakeholders. These objectives are not contradictory, but they're not consistent. Is there a difference between a key objective and a primary objective? Perhaps the former a subcomponent of the latter? The objective on Line 88 specifies Arctic settings, but the objective on Line 113 expands this to the Arctic and Subarctic. Table 2 confirms the study includes both Arctic and Subarctic locations. It may seem like nit-picking but keeping the objective wording consistent and referencing it consistently throughout the paper will be immensely helpful to the reader. I think Line 449 objective is the strongest in terms of clarity, and should be provided in the introduction (I would recommend specifying Arctic/Subarctic coastal environments in the wording of this). If the key objective referenced in Line 113 is indeed a subcomponent of an overall objective, it should be clarified as such, and also provided in the Introduction. The authors could refer to them as Primary Objective (PO) and Specific Objectives (SO1, SO2, etc.), and refer to them throughout the paper using these abbreviations.

*We agree that there was a fragmented statement of the objective in the original submission. In the revised manuscript, we follow the reviewer's suggestions and state the primary objective consistently (from the original Lines 449-451) in the Abstract, the Introduction and the Conclusion: "The primary objective of this study is to use locally-based metrics to construct indicators of break-up and freeze-up in the Arctic/Subarctic coastal environment".*

Line 118: Is it possible to clarify what "close" is? I understand the authors can't use adjacent passive microwave cells due to land contamination, but maybe the next cell over? In which case since the spatial resolution is 25km would it be fair to say the authors chose cells beyond 25km from the coastline? Phrasing it this way is consistent with how the cells appear to be distributed in Figure 1. However it is interesting to see that the cells are much further away from the coastline in the Chukchi Sea compared to the other study locations. What informed this decision?

Provide a sentence or two explaining the exact considerations that went into cell selection beyond saying the cells are close to the coastline.

*The reviewer's interpretation is correct: Because cells adjacent to the coast were excluded, the selected grid cells satisfied the criterion that they were the cells closest to the coast but centered at least 25 km from the coast. The revised text explicitly states this criterion. The Chukchi Sea grid cells represent an exception chosen specifically to allow for comparison with a location farther from the coast (offshore from the continental shelf and unaffected by landfast ice), as discussed in Section 2.*

Line 127: No need for the apostrophe in "weeks"

*Apostrophe has been removed.*

Line 154: Typo in "Indigenous community"

*Typo has been corrected.*

Line 157: An example of why elaboration on why the authors chose the cell locations they did would be helpful (per my feedback on line 118) The significance of St. Lawrence Island in the Bering Sea is, per Table 1, due the location of indigenous communities. However, because the communities of Savoonga and Gambell are situated on the Island's northern coasts, I am curious why the authors chose cells to the south of the island for this study area? Does it pertain to hunting locations for these communities? These cells are also proximate to the southern polynya that keeps the ocean beyond the island's barrier islands relatively ice-free. Did this factor into the authors decision to select this spot? Providing rationale for cell location choice could even be included as an additional column in Table 1, if the rationale varies by study location.

*The near-shore locations were selected to provide a geographically distributed set of sites at which sea ice affects coastal users. The affected users were chosen to be a mix of coastal villages, industry and military sites, as indicated in the discussion of Table 1. There is admittedly some subjectivity in the selection of the sites, and we now state this in the revised text, but we believe that the geographical distribution provides a defensible rationale for the selection. The revised text emphasizes that geographical distribution and impacted human activities were the primary considerations in the site selection.*

*A new paragraphs has been added (p. 12) specifically to address the reviewer's comments with St. Lawrence island and the communities of Savoonga and Gambell. We have added text that addresses the review comments on p. 9. We have also added the following new text (p. 12/bottom – 13/top) to explain the St. Lawrence grid cell selection:: The grid cell selections for*

*St. Lawrence Island and the Chukchi Sea deserves special comment. The grid cells off St. Lawrence Island were chosen to reflect timing and location of subsistence harvests by the communities of Gambell and Savoonga. Because of extensive ice coverage, including landfast ice, north and northwest of the island, both communities traditionally conduct bowhead whale harvests at hunting camps on the south side of the island once spring ice break-up is underway (Noongwook et al., 2007). These sites also reflect the seasonal migration of whales in waters south of the island with the seasonal retreat of the ice cover (Noongwook et al., 2007), modulated somewhat by the presence of a polynya south and southwest of the island (Krupnik et al., 2010; Noongwook et al., 2007). Traditional walrus harvest practices on St. Lawrence Island await the tail end of the bowhead whale hunt (Kapsch et al., 2010), with timing of spring ice break-up south of the island as the driving factor. These practices motivated our selection of grid cells southeast of the island. As shown later (Section 4), landfast ice is confined to the northern coastal region of St. Lawrence Island – consistent with the frequent presence of the polynya south of the island.  In the case of the Chukchi Sea, the grid cells are indeed farther from the coast than for the other sites; the locations were intentionally selected to be farther offshore in order to provide a non-coastal counter-example to the other sites, all of which are adjacent to a coast.*

Line 160: Is it possible to re-cite so we know exactly which of the cited literature is relevant to break up and freeze up metrics?

*We have added references to Markus et al. (2009), Johnson and Eicken (2016), Bliss and Anderson (2018), Peng et al. (2018), Bliss et al. (2019) and Smith and Jahn (2019).*

Line 170: Is it possible to include the cell locations in Figure 2? Certain regions (e.g. Baffin Bay, Bering Sea) are very large, and the authors are looking at very specific areas within them. Perhaps adding them as bright red/yellow stars denoting general cell location could help readers to connect Figure 1 with Figure 2. Also, I notice the Canadian Archipelago (9) is missing in the figure caption. Please fix this.

*We have made a revised version of this figure (now Figure 1) with black dots at the locations of the 10 sites on which the study focuses.  We have also added the site locations with place-names to Figures 3 and 7 from the original submission, as well as the new Figure 15 showing the landfast ice distribution. This suggestion is much appreciated because the broad geographical distribution is now readily apparent in the revised figures.*

Line 288: Another Study Objective that can be referenced using consistent terminology (see my feedback regarding Line 113).

*As noted in the earlier response (Line 113), we revised the stated objectives for consistency. The statement here (previously Line 288) has been reworded to conform to the statement in Lines 449-451 of the original manuscript (and also the revised abstract).*

Line 302: St. Lawrence Island is generally encircled by landfast ice between January and May/June, and can exceed 25 kilometers off the northern coast. However, the cells are located south of the island, where the presence of a latent heat polynya inhibits the seaward advancement of the landfast ice edge beyond the barrier islands. This polynya also keeps the area relatively free of drift ice. At least one of the three cells are within the polynya boundary, which generally does not extend east of the island's southeast cape. Was this a consideration in the interpretation of the results? Rather than due to being free of landfast ice, earlier break up could be detected in this region because the wind-driven advection of sea ice out of the cell's locations compared to other areas in the MASIE region, which is the Bering Sea in its entirety. Again, given the size of the MASIE subregions, I'm not sure why this particular area for the Bering Sea was selected for cell placement unless the authors provide information regarding what factors were considered beyond proximity to the coastline.

*The grid cells off St. Lawrence Island were chosen to reflect timing and location of subsistence harvests by the communities of Gambell and Savoonga. Because of extensive ice coverage, including landfast ice, north and northwest of the island, both communities traditionally conduct bowhead whale harvests at hunting camps on the south side of the island once spring ice break-up is underway (Noongwook et al., 2007). These sites also reflect the seasonal migration of whales in waters south of the island with the seasonal retreat of the ice cover (Noongwook et al., 2007), modulated somewhat by the presence of a polynya south and southwest of the island (Krupnik et al., 2010; Noongwook et al., 2007). Traditional walrus harvest practices on St. Lawrence Island await the tail end of the bowhead whale hunt (Kapsch et al., 2010), with timing of spring ice break-up south of the island as the driving factor. These practices motivated our selection of grid cells southeast of the island. We have revised Figure 1 to show the location of landfast ice in the vicinity of each coastal location, including St. Lawrence Island, where the new figure shows landfast ice confined to the northern coastal region – consistent with the frequent presence of the polynya south of the island, as noted by the reviewer.*

*New references:*

*Kapsch, M.L., Eicken, H. and Robards, M., 2010. Sea ice distribution and ice use by indigenous walrus hunters on St. Lawrence Island, Alaska. In SIKU: Knowing Our Ice (pp. 115-144). Springer, Dordrecht.*

*Krupnik, I., Apangalook, L. and Apangalook, P., 2010. "It's Cold, but Not Cold Enough": Observing Ice and Climate Change in Gambell, Alaska, in IPY 2007–2008 and Beyond. In SIKU: Knowing Our Ice (pp. 81-114). Springer, Dordrecht.*

*Noongwook, G., Native Village of Savoonga, Native Village of Gambell, Huntington, H.P. and George, J.C., 2007. Traditional knowledge of the bowhead whale (Balaena mysticetus) around St. Lawrence Island, Alaska. Arctic, pp.47-54.*

Line 304: Interpretation of results is generally best confined to the Discussion section. That being said, the wording is not clear in how landfast ice is a key determinant of the timing of break up. I assume the authors are saying the break up start dates are later in the coastal regions than the MASIE regions because landfast ice generally persists later than areas dominated by drift ice? In which case this needs to be clarified with more specific wording. "Landfast ice generally persists in coastal areas longer than drift ice at the end of the season, and is therefore a key determinant in the timing of later break up onset relative to the broader sector of the seasonal ice zone". Something like that which is more clear. However, unique geography of each region (e.g. the polynya off of St. Lawrence Island) complicates speaking in such general terms about the role landfast ice plays in comparatively later break up detection in coastal regions versus MASIE regions. I recommend some sort of effort to validate that this is the case. This can be accomplished by referencing concurrent datasets such as ice charts or satellite imagery, or even cited literature with qualitative descriptions of where/when landfast ice is located relative to cell placement. It is insufficient to say whether or not the general area — not even the specific area occupied by the cells — is prone to landfast ice build up.

*The need for more specificity about landfast ice is a valid criticism of the original submission, so we have augmented this section with additional discussion (placed in a new Discussion section, as suggested) and a new figure (Figure 15) showing the median and maximum spatial extent of landfast ice during June (the middle of the break-up period). This map is based on the charts of the U.S. National Ice Center for the 35 years ending 2007. We have superimposed on this map the locations of our ten local sites, thereby providing a basis for our assessment of the potential role of landfast ice in delaying the break-up. In addition, we have revised Figure 1 to include landfast ice masks for each panel. As explained in the text, the landfast dataset contains an offset of the coastline relative to the NSIDC passive microwave dataset, but the area covered by landfast ice near each of our sites is apparent from the new Figure 1.*

Line 323: Regarding Figure 8, when the timing of sea ice events are being studied, especially onset and break up, some studies will consider September 1st of the previous year to be the Day 1, and August 31 of the following year to be day 365 (e.g. September 1st 1996 - August 31 1997). Because Break Ups in this figure are generally between Days 60 - 180, it appears January 1st is considered Day 1. This is fine, but the authors may want to specify this in the Figure 8 caption, to avoid confusion for any readers expecting September 1 to be Day 1. Also, I notice the y-axis scales are not standardized. Was this done intentionally? I can see how standardizing the y axis to the Sabetta / Kara Sea, which has the largest spread of values, may make it hard to interpret plots with a tighter spread of y axis values (e.g. Tiksi / Laptev Sea) by pushing the points closer together. If it is possible to standardize the y-axis, I would recommend it. However if this makes it difficult to interpret the plots, it's okay to leave the y-axis as is and make mention of this in the caption. Also, it is difficult to see the standard error portions of the MASIE region

with the current color. Can the authors choose a darker shade of pink so that the SE stands out better?

*The revised caption clarifies that the day number is the Day-of-the-Year (Julian day number). We also added to the caption a statement that the scales on the y-axes vary regionally in order to make the various panels more reader-friendly.  We added to the caption a statement that the slopes and significance levels are now provided in the Supplementary Material, Tables S2 and S3. Finally, we have darkened the shading in each panel in order to make the standard error more apparent.*

Line 330: Since this figure is showing the same thing as Figure 8 for the break up dates and does not require a caption, it would be better to combine them into a single panel of figures, 8a, and 8b, (8c, and 8d for freeze up) with one caption. This may be disregarded if the figure becomes too big. However the lack of captions for Figures 9, 10, and 11 due to their similarity makes me think it would be better to just combine them.

*We tried a layout of the figure as suggested by the reviewer. However, with 40 panels, it was decidedly reader-unfriendly, and it would not fit on a single journal page (at least legibly).  As a compromise, we have expanded the headers to highlight the differences among the figures, and we expanded the captions of Figures 9-11 so that the reader has the essential information below each figure.*

Line 397: I am wondering why the authors chose to put the x axis ticks in the middle of the graph instead of the bottom? Please correct.

*Figure 13 has been redrafted so that the x-axis scale is at the bottom of each panel.*

Line 443: I did notice a lot of results interpretation included in the results section, and was expecting to see this in a Discussion section. Why is this section omitted, and we go right from Results to Conclusion?

*The revision includes a separate section (Section 4) entitled Discussion.  This new section contains the interpretation sections that were previously in Section 3, and it also includes an expanded discussion of the role of landfast ice based on the new landfast ice figure  (Figure 15). It also includes our response to the reviewer's comment on Line 304 and Line 457 (see following response)..*

Line 457: The authors are saying later break ups in the study areas are due to the lag of landfast ice break up. However, given the relatively small area these cells occupy compared to the larger region the authors say are "known to have extensive landfast ice" makes this connection weak. Landfast ice can be highly spatially heterogenous, even in regions that are known to have large extents of cover. I recommend the authors find a way to validate that the later sea ice break up is indeed due to landfast ice. This may be accomplished with existing landfast ice datasets. The Canadian Ice Service provides sea ice charts for the northern coast of Alaska, the Canadian Archipelago, Hudson Bay, and Baffin Bay, including the timing and location of landfast ice regions. Simply taking a sample of later break ups and confirming the cells are occupied by landfast ice during this time would strengthen the connection between later break ups and landfast ice.

*As noted in the response to the reviewer's comment on Line 304, we added substantiation of the role of landfast ice in a new Discussion section. This section includes a new figure showing the spatial extent of landfast ice during June (the middle of the break-up period) in the form of median and maximum landfast ice extent. We have superimposed on this map the locations of our local sites, thereby enabling a demonstration of the correspondence between delayed break-up and the presence of landfast ice and providing a basis for our assessment of a role of landfast ice in delaying the break-up. This new material supports the previous statement about the role of landfast ice in most instances, although there are a few exceptions that we point out in the text.*

Line 458: The main benefit of including a discussion section is to interpret and contextualize the results with the broader body of scientific literature. Right now, there's no discussion section, and results interpretation in the conclusion section does not cite any literature where results are interpreted. There is plenty of literature explaining how shallow bathymetry and freshwater inputs facilitate earlier sea ice freeze up in coastal zones. These results are consistent with the findings in that literature, why is it not cited? The omission of a discussion section and lack of literature cited in results interpretation prevents readers from connecting this work with the broader body of scientific knowledge. It weakens this manuscript from serving as the basis for future research. This is a significant problem that needs to be addressed before this manuscript is suitable for publication. Any writing interpreting the findings in the Results section should be moved to a discussion section, with cited literature throughout.

*We agree with this assessment, and our response is threefold: (1) the addition of a new Discussion section (Section 4) in which the interpretations formerly under "Results" are placed, and (2) an expansion of the material on landfast ice and its relation to delayed break-up, and (3) the addition of new references to previous studies of landfast ice and its role in the timing of break-up (Petriich et al., 2014; Mahoney et al., 2007; Mahoney et al., 2014; Yu et al., 2014; Hoskova et al., 2021; Cooley et al., 2020). These additional references enable us to better place our results into the context of prior work.*

**Reviewer 2**

**RC2**: ['Comment on tc-2022-21'](), Anonymous Referee #2, 11 Mar 2022

Review of Walsh et al., "Sea ice break-up and freeze-up indicators for users of the Arctic coastal environment," submitted for publication in The Cryosphere.

This manuscript describes a nice study focused on the difference in ice break-up and freeze-up dates between near-coastal locations vs larger scale regional averages. The paper is generally well-written and the figures are clear. The results are convincing and will have an impact on how such analyses are used for coastal sea ice science. I do have some questions and comments, so I recommend for publication with major revisions.

Lines 63-64: As noted below, Bliss does not use this type of definition. So there are a variety of methods.

*We have modified the text to say "Sea ice concentration thresholds have been used in various studies to determine the dates of sea ice opening, retreat, advance and closing.(Markus et al., 2009; Johnson and Eicken, 2016; Bliss and Anderson, 2018; Peng et al., 2018; Bliss et al., 2019; Smith and Jahn, 2019). For example, Bliss et al. (2019) define dates of opening and retreat as, respectively, the last days on which the sea ice concentration drops below 80% and 15% before the summer minimum. Corresponding metrics are used by Bliss et al. for the dates of advance and closing."*

Lines 77-78: The 25 km PMW products provide the longest time record. However, we have at least 20 years of higher resolution AMSR data as well. Do you think your analysis would differ if you used that? I suggest you acknowledge that higher resolution PMW data are available and that they might be uniquely useful for examining coastal science.

*The higher-resolution AMSR data would resolve the near-shore sea ice distribution that is a focus of this study. We have added the following after lines 77-78 (original line numbers): "While higher-resolution datasets permitting finer resolution of coastal sea ice are available from sensors such as AMSR (Advanced Microwave Scanning Radiometer), the record lengths are sufficiently shorter (about 20 years for AMSR) that trend analysis are limited by a reliance on such products. Trend analysis is one of the main components of the present study."*

Line 94: You've got a typo here. NSIDC-0051 is the NASA Team data set. You want to write NSIDC-G02202, the CDR data set. And also note that there is a Version 4 available: was this not available when you started this work?

*We did use the data from NSIDC-0051. We have deleted the reference to the NOAA/NSIDC Climate Data Record. The documentation for this dataset ([https://nsidc.org/data/nsidc-0051](https://nsidc.org/data/nsidc-0051))*

*points to the use of the NASA Team algorithm. The analysis described here was initiated prior to the availability of Version 4 of NSIDC-G02202.*

Line 107: "Examples include…" I suggest you clearly write out all modifications from the previous algorithm, so your methods can be potentially reproduced by others.

*This comment is similar to one from Reviewer 1. In response, we have added a paragraph (following the previous Line 112) summarizing the changes in each indicator relative to the criteria used by Johnson and Eicken (2016). In brief, the following modifications to the K&E criteria were incorporated into the algorithms:*

*Break-up start:*
*- minimum sic threshold created at 15% (vs. last day exceeding Jan-Feb mean minus 2σ)*
*- undefined if average summer sic > 40% (vs. no such criterion in J&E)*
*- undefined if subsequent breakup end date not defined*

*Break-up end:*
*- first time sic below threshold for 2 weeks instead of last day below threshold*
  *(vs. last exceeding larger of Aug-Sep mean or 15%)*
*- minimum threshold 50% instead of 15%*
*- undefined if break-up start not defined*

*Freeze-up start:*
*- first day on which sic exceeds Aug-Sep average by 1σ (as in J&E)*
*- undefined if mean summer sic > 25%*
*- undefined if subsequent freeze-up end not defined*

*Freeze-up end:*
*- first time sic above threshold for following 2 weeks instead of first day above threshold*
  *(threshold is Jan-Feb average minus 10%, as in J&E)*
*- thresholds imposed: Minimum (15%) and maximum (50%)*
*- undefined if sic always exceeds threshold*

Lines 110-111: I don't think v4 of the CDR has missing days.

*See response to comment on Line 94. We did not use the CDR.*

Line 111: "a spatial and then temporal smoothing" Please provide all details.

*The following has been added in the revised manuscript: "The daily sea ice concentration values were spatially smoothed using a generic mean filter with a square footprint of 3 x 3 grid cells. This data was then temporally smoothed three times in sequence using a Hann window"*

Table 1: There are a large number of seemingly arbitrary numbers here. Why 25%, or 40%, or 50%? Why two standard deviations, or one standard deviation? Why minus 10%? First, why did you pick these, and second, what if you vary these numbers?

*The 10% threshold is based on prior work (Johnson and Eicken, 2016) in which sensitivities were explored. The 25%, 40% and 50% thresholds in Table 1 were arrived at by testing various values and selecting values that maximized the number of years with break-up and freeze-up dates defined. The selected values were those that generally maximized the number of such years across the various coastal locations and MASIE regions. We have added this information to the text (paragraph preceding Table 1).*

* You are providing the lat/lon of these coastal communities/stations. Why? They are not used anywhere. Instead, perhaps you can provide the exact grid cell numbers of the 3 cells for each location.

*In the interest of a general readership (one that does not work specifically with the NSIDC passive microwave products), we believe that the latitude and longitude will be more useful to readers than the grid cell coordinates. The latter have the disadvantage that they will vary with the sea ice dataset and its resolution, while the latitude and longitude are site-specific and fixed.*

* Mestersvig seems to be decommissioned, as far as I can tell. Why did you choose this?

*While the permanent military function at Mestersvig has been discontinued, the site has been used as recently as 2015 for military exercises to test the responsiveness of Denmark's military and the cold-weather functionality of its equipment. Since the airport (with its 1800 meter airstrip) and other infrastructure, further use of this site is possible, if not probable.*

Figure 1:

* Could you rotate all panels so that north is upward, as is usual?

*In the revised version of this figure, we have rotated the panels for consistent north-south orientation and have added a northward-pointing "N" arrow in each panel.*

\* "Bering Strait coast" is an odd name for this location. Really it is "NE Chukchi Sea," right? Also, these 3 grid cells are farther away from the coast than the other locations. Why?

*This location is actually in the southern Chukchi Sea, as shown in our revised Figure 1, which (per the suggestion of Reviewer 1) shows all the locations listed in Table 3. We have changed the name of this site in the Table (and text) to simply "Chukchi Sea". This location is indeed farther from the coast than the other sites, and the location was intentionally selected to be farther offshore in order to provide a non-coastal counter-example to the other sites, all of which are adjacent to a coast. We added this information to the text (end of the 2nd paragraph following Figure 1).*

\* I suggest adding distance or lat/lon markers along the horizontal and vertical axes of each panel.

*As noted above (2nd preceding bullet point), we have reoriented all panels in Figure 1 so that north is consistently at the top of each panel. We have also added a distance scale indicator to each panel.*

More on Figure 1: It took me a few minutes to understand the basic strategy here, so I suggest that you make this clearer. You show these coastal locations with red dots, but really they are never used. However, this distracted me and I thought you were going to compare coastal station information with these three grid cells. No, in fact you are using these 3 grid cells as proxies for coastal conditions. I suggest you say this explicitly. And then I am wondering:

*We have revised the caption to state explicitly that the red dots (black dots in revised version) represent the actual locations of coastal communities, while the offshore squares represent the grid cells of the passive-microwave-derived sea ice concentrations used in computing the break-up and freeze-up metrics.*

1) Why not choose SIC cells right at the coast? Yes, there could be coastal contamination, but I think the CDR has eliminated or much reduced this. At least you should check this.

*In our examination of the coastal regions in NSIDC-0051, we found that there is indeed still some contamination present in grid cells adjacent to the coastline. For this reason, we chose grid cells that were as close as possible to the coastline without being at the coast (i.e., we required that there be a one-cell offshore displacement of any selected grid cell).*

2) Why not validate these CDR SIC values with coastal information when available? EG Utqiagvik has a sea ice radar, doesn't it?

*There is a sea ice radar at Utqiaġvik, but its range is less than 25 km so it does not "see" beyond the grid cell immediately adjacent to the coast.  The northern Alaska coast is an example of a region in which coastal contamination led to non-zero ice concentrations at coastal grid cells when sea ice was not present (e.g., mid-September of years in the final decade of the dataset).*

3) What if you chose a trio of CDR SIC cells that were at another location near the coast within a MASIE region? Do you think you would get a different result? I suggest that you take one or two MASIE regions and test this, ie take 3 different near-coastal locations within a MASIE region and run your analysis. How different are your results?

*In response to this comment, we actually experimented with the grid cell selections at Utqiaġvik and Sabetta.  As discussed in a new paragraph (p. 9-10), the mean values of most of the metrics changed little (by no more than 1.1 day) when the near-coastal grid cells were shifted offshore by one pixel. The one exception was the break-up at Utqiagvik, where the one-pixel offshore offset resulted in an earlier break-up start by 3.3 days (consistent with a delay of break-up by landfast ice along the coast) and a later break-up end by 2.9 days.*

4) You interpret many results in the context of landfast ice. I think you need to show that the CDR SIC cells you have chosen (because they are offshore) are actually in the landfast ice zone at each location. EG you could add a typical landfast ice contour (when it's there) to each panel of Figure 1. Or better, some kind of interannual min/max measure of landfast ice extent relative to the cells.

*This concern was also raised by Reviewer 1.  In response, we are augmenting this section with additional discussion (placed in a new Discussion section, as suggested by Reviewer 1) and a new figure (Figure 15) showing the maximum spatial extent of landfast ice during June (the middle of the break-up period).  These maps are based on the charts of the U.S. National Ice Center for the 35 years ending 2007. We have superimposed on this map the locations of our coastal sites, thereby providing a basis for our assessment of a role of landfast ice in delaying the break-up. In addition, each panel of the revised Figure 1 shows the median/maximum June extent of landfast ice.*

Line 163: Typo: "Figure 1" -> "Figure 2"

*Corrected in revision.*

Lines 165-167: Why did you include MASIE regions that are not near your 10 locations, eg "Central Arctic?"

*The original text was intended to list the entire set of Arctic and subarctic regions in Figure 2. In the revision, we have omitted from these lines the regions that we do not use in this study (Regions 6, 9 and 11). We have also replaced Figure 2 with a new version that omits the region numbers and shows only the region names.*

Line 193: Typo: "Table 2" -> "Table 1"

*Corrected in revision.*

Figure 3: It might be useful to remind the reader in this caption that break-up start only exists when there's break-up end, and same for freeze-up, thus only two panels are needed in this figure.

*This reminder has been added to the caption of Figure 3.*

Lines 213-216: This is interesting; I think Bliss vs J&E is in some ways analogous to NASA Team vs NASA Bootstrap. IE Team uses fixed tie points, while Bootstrap uses "dynamic" ie time/space varying tie points. There is value in each method (although generally the latter seems to be more accurate).

*We have added a statement pointing to this analogy between the algorithms and the J&E vs. Bliss differences (end of first paragraph after Figure 3).*

Figure 4: I had to look up "violin plots" but there is a Wikipedia page so ok. Still, it might be nice to write one sentence explaining what this is. Also:

*We have expanded the caption of Figure 4 to explain what a violin plot actually is.*

* What are the black strips within each histogram? Are they gaps where there's no data for that day of year?

*The thin black lines represent the observations themselves, i.e., the indicator dates for each year. The black strips are clusters of lines representing groups of similar values in the distribution,*

*i.e., in this case they represent a narrow range of dates in which multiple years had their break-up or freeze-up dates.  We have added this information to the caption of Figure 4.*

\* Perhaps you want to mark the mode or mean on each histogram, and write it as text?

*We have added a new table (Table S1) listing the mode (peak of distribution) of the dates in the violin plots   For each metric and location, the table includes the corresponding modal values from J&E and Bliss et al.*

Figure 5: I suggest writing in text the two slopes in each panel within each panel (or you could create a separate table). And this goes for all subsequent similar figures with trend lines. Perhaps one big new table with all slopes for all such figures together?

*Since legible font would clutter the 12-panel figures, we will have added two new tables in the Supplementary Material (Tables S2 and S3) listing the slopes and their significance levels for all the regression lines in Figures 5-6 Figures 8-11.*

Line 377: Typo: "…and negative…" ie add "and"

*"and" has been inserted in revision.*

Line 387: Typo: "Figure 12" -> "Figure 13"

*Corrected to "Figure 13" in revision.*

Line 411: Here is a main result: The variance at three grid cells within a large region is higher than for the regional mean. This is useful in the context of coastal science, and it is very reasonable i.e., intuitive. I might propose that it is too reasonable, i.e., isn't this expected that variance increases when you focus on a small subregion? Actually, this could be true in some cases and the opposite might be true in others (eg if you picked three grid cells in a "boring" place with low variance). What if you took a MASIE region and made a map of some factor analysis parameter that would illustrate this? Would it show higher variance near the coast and lower variance toward the perennial ice pack? This seems important for you in order to really show that your result is not so obvious.

*We had some difficulty in interpreting this comment, but we tried to address it by performing an additional factor analysis on the ice concentrations within a MASIE region (Figure 1's Chukchi Sea region). The strongest factor loadings were found in the south, where more years experience freeze-up and break-up than in the north. As the reviewer notes, this is not surprising because the variance is greater in the south than in the north. We have added a statement in the text after the statement beginning Line 411 (original submission), pointing out that the difference in variance indeed contributes to the difference in variance explained for the local and regional indicators. However, we do not find it so obvious that the percentage of variance explained by the first factor should be greater for the more localized metrics. We now state this in the text (p. 32, middle).*

---

## Author Response (AR2)

**Responses to Reviewer 2 (2ⁿᵈ round)** -- *responses in italics,* reviewer comments in normal font)

*First of all, we thank Reviewer 2 for the helpful comments and the careful reading of the revised manuscript. We have revised the manuscript to address essentially all the comments, as described in the point-by-point responses below.*

Review of revised manuscript, "Sea ice break-up and freeze-up indicators for users of the Arctic coastal environment" submitted for publication in The Cryosphere.

This revised manuscript addresses many of my original comments, thanks for this. However, the text could still use some grammar/spelling corrections. The file with track changes lacks line numbers which would have made my reviewer's job easier. There are still missing explanations that should go in the Data and Methods section. My main comment is that the landfast ice influence is still unclear to me, and I feel that it requires further analysis if this is to remain a key finding. I recommend another round of major revisions.

Figure 1 is much improved, thanks. Unfortunately, unless I am misinterpreting it, it still falls short in supporting a key finding of this work, namely that landfast ice strongly influences seasonal ice breakup & freezing timing. Most of the trios of sea ice concentration cells in Figure 1b lie outside of the median landfast zone, which is a problem for your finding, isn't it? Further, while in theory it's a good idea to partition your locations into those that are or are not strongly influenced by landfast ice, e.g. on page 23:

*According to Figure 2 (formerly Figure 1), The selected grid cells for only three of the ten coastal sites lie outside the landfast ice: Chukchi Sea, St. Lawrence Island, and Utqiagvik. As noted in the text, the Chukchi Sea site was intentionally chosen to be well offshore of the coast in order to provide a "control" site. The St. Lawrence Island grid cells were also chosen intentionally to be outside the landfast ice area for reasons related to stakeholder uses of the offshore waters, as described on p. 15. For Utqiagvik, we summarize on p. 11 (bottom) – p. 12 (top) the results of a sensitivity analysis that showed there was negligible change in the computed metrics when the three pixels were given a one-cell displacement perpendicular to the coastline.*

"The break-up start date at the coast is later than for the MASIE regions for Prudhoe (Beaufort Sea), Utqiaġvik (Chukchi Sea), Tiksi (Laptev Sea), and both Canadian locations: Churchill (Hudson Bay) and Clyde River (Baffin Bay). These sites are all Arctic coastal locations at which varying extents of landfast ice are present. By contrast, the coastal locations have earlier break-up start dates (relative to their corresponding MASIE regions) at St. Lawrence Island (Bering Sea), Mestersvig (Greenland Sea) and the Bering Strait (Chukchi Sea). These locations are less prone to experience a buildup of landfast ice during the winter."
your interpretation of Figure 1 is puzzling, e.g. (1) The trio of sea ice concentration cells near Utqiagvik lie outside of both median and maximum landfast ice shown in Figure 1, so "varying extents of landfast ice" are NOT present there, and (2) The trio near Mestersvig lie within the maximum landfast ice zone: why is it included in the "less prone" group? I also find your new Discussion section overly long, out of place (I think a shorter version should reside in the Introduction), and not adequately tied to your specific results. Thus I think that more analysis is

required to adequately support your hypothesis of strong landfast ice influence. For example, you could show a multi-year mean daily or monthly time series of landfast ice concentration or extent averaged over each sea ice concentration cell trio, with vertical bars showing interannual variability using standard deviation or quartiles, etc. This would provide information on the seasonal timing of when landfast ice comes and goes, in addition to supporting "less" or "more" "prone to landfast ice" statements.

*We have responded in several ways to this general comment.  First, we have shortened the Discussion section, both by moving some material to the Introduction and by eliminating some non-essential text.  In particular, the background material on landfast ice and the pan-Arctic map of landfast ice distribution (formerly Figure 15) have been moved to Section 1.  Second, in order to more substantively address the role of fast ice, we have added a new table keyed to the impact of landfast ice.  This new table (Table 5) lists the sites in approximate order of decreasing lag between break-up at the local sites and the broader MASIE regions.  The table also indicates in Column 2 whether landfast ice appears to play a role in the lag.  The accompanying text clarifies the role of landfast ice at a few sites questioned by the reviewer (e.g., Mestersvig and St. Lawrence Island, Chukchii Sea).  We note that the correspondence between landfast ice and the lag of break-up is evident but not perfect, and that the exceptions (Pevek, Sabetta) provide opportunities for further research to test the hypothesis presented in our discussion.*

I don't think a "track changes" file should include your internal comments.

*We have removed all internal comments from this change-tracked revision (Revision #2).*

Page 4:
1) In the abstract, the primary objective "is;" here it "was." Probably "is" is best for both.

*Changed as suggested – see revised abstract.*

2) "A subcomponent of this overall objective" -> "A secondary objective"

*Changed as suggested – see bottom paragraph on p. 6.*

Page 5:
1) "In the construction of this dataset, the NASA Team algorithm (Cavalieri et al., 1984) and the NASA Bootstrap algorithm (Comiso et al., 1986) were used to process the microwave brightness
temperatures into a consistent time series of daily sea ice concentrations." No, this is incorrect. NSIDC 0051 is pure NASA Team. This sentence refers to the CDR which evidently you did not use.

*Text has been corrected to cite only the NASA Team algorithm.*

 2) "Prior to applying these definitions," What definitions?

*Reference to "these definitions" has been removed.*

3) What is a "generic mean filter?" Do you mean a boxcar filter?

*Yes, we used a boxcar filter and now state this explicitly.*

Page 6:
1) "based on past studies and subsequent sensitivity tests" -> based on past sensitivity tests
2) "at by testing various values and selecting values" -> at by selecting values
3) "with break-up and defined freeze-up dates" -> with break-up and freeze-up dates

*(1)-(3) have all been corrected.*

4) "had the best agreement with years of indigenous observations" This needs a much more extensive explanation, or perhaps reference to a previous publication.

*p. 7 (bottom) – p. 8 (top), we now provide two supporting references in this statement: "The definitions build on those used by Johnson and Eicken (2016; hereafter denoted as J&E), which were informed by Indigenous experts' observations of ice use and ice hazards in coastal Alaska, and relate to planning and decision-making at the community-scale (Eicken et al., 2014)".*

5) "The selected values were those that generally maximized the number of such years across the various coastal locations and MASIE regions." I think you've already said this and so this can be cut.

*Repetitive statement has been deleted.*

Page 9: "the relevance of local sea ice to uses by communities" ?? not sure what is meant here

*A clarifying statement has been added, with examples (p. 11, top paragraph).*

Page 12:
1) "innermost extent of the landfast ice does not always coincide with the coastline." You should include a sentence that tells the reader that you are assuming that this is always wrong. IE landfast ice always DOES coincide with the coastline, is that correct?

*Yes, that is correct. We have added this clarification to the text on p. 14, bottom paragraph.*

2) What is the "nearshore edge?" You just implied (and should make explicit) that there is no "nearshore edge" ie that landfast ice coincides with the coastline, correct? So I am confused.

*We have reworded this to refer to the "inner boundary" of landfast ice (p. 14, bottom).*

Page 12:
Cut "(8)" near the bottom

*(8) has been removed.*

Page 14:
"The computation of the indicators was dome for the tem local" "m" -> "n" x 2

*Corrected to "ten" (p. 17, line 4).*

Page 17:
1) "dynamic -> "dynamic"

*Corrected.*

2) "corresponding indicators used by Bliss et al. (2019)" This material should go in the Methods section. Did you make any modifications to the Bliss method?

*We have moved summary of the the Bliss et al. (2019) algorithm to Section 2 (p. 8, 2$^{nd}$ para.). On p. 19, where we introduce the comparison to Bliss et al. by simply stating the two concentration thresholds used by Bliss et al.*

Page 20:
"a sufficient number of years" What is "sufficient?" This should be in the Methods section.

*We have revised this statement to say that the Central Arctic metric was defined for "fewer than half (p. 22, bottom).*

Page 22:
"assess the relationship between the local indicators and those for the broader MASIE regions containing the coastal locations" I think this is useful, but some caveats are in order. Your diagnostic parameters were tuned to coastal conditions, and even vetted by coastal users, yes? So, you should caution the reader that applying them to a large area that includes far-offshore regions may be a misapplication. IE your method may likely be "better" for coastal applications but it is not necessarily better (and could be worse) for either regional or pan-Arctic studies.

*The reviewer makes a good point here, so we added the caveat about the local vs. MASIE indicators (p. 24, bottom – p. 25, top).*

Page 26:
"These are regions in which it is common for ice to form along the coast in autumn, with the ice edge advancing offshore to meet the expanding main ice pack as freeze-up progresses." Where is your proof of this? Perhaps my suggestion of moving your Discussion material to the Introduction could help solve this problem.

*In order to support this point about the progression of freeze-up, we added a new figure (Figure 12) containing examples. We prefer to keep it in Section 3 because it fits the flow of that section as we transition from the break-up to freeze-up indicators.*

Page 29:
1) "middle months of the break-up and freeze-up seasons (June and November, respectively)" These seasons should be defined in the Methods section.

*In Section 2 (Methods), we have added text clarifying the definitions of the break-up and freeze-up seasons, including our choice of June and November as representative months. See p. 8, middle paragraph.*

2) "Pacific hemisphere" This should be defined in the Methods section. I notice that Tiksi is included.

*We have added the definition of the Pacific Hemisphere on p, 31 (near bottom of page).*

Page 35:
1) As noted above, I suggest shortening this material and moving it to the Introduction.

*As noted above, we have shortened this section and moved parts of it (including the background information on fast ice) to Section 1 (p. 5-6).*

2) "lengthening of the open water season" Previously you discussed shortening of the ice season; you did not discuss lengthening of the open water season.

*The correspondence between a "lengthening open water season" and a "shortening of the sea ice season" is not stated explicitly on p. 5 (start of bottom paragraph).*

3) "The timing of break-up and freeze-up relates to the proximity to the coast" I cannot find where you showed this.

*We have reworded this statement to say that the MASIE regions are centered farther from shore than the coastal grid cells (p. 37, first para.).*

Page 36:
1) "In most cases, these differences can be related to the presence of landfast ice" I don't think you've proven this.

*We have attempted to support this statement by adding a new Table (Table 5) and discussion of it (p. 37-39). This new table lists the sites in approximate order of decreasing lag between break-up at the local sites and the broader MASIE regions. The accompanying text clarifies the role of landfast ice at a few sites questioned by the reviewer (e.g., Mestersvig and St. Lawrence Island, Chukchii Sea). We note that the correspondence between landfast ice and the lag of break-up is evident but not perfect, and that the exceptions (Pevek, Sabetta) provide opportunities for further research to test the hypothesis presented in our discussion.*

2) Is Figure 15 necessary? I suggest not: I think Figure 1 is sufficient.

*In view of the expanded discussion of landfast ice in the paper, together with the limited geographic areas shown in the old Figure 1 (now Figure 2), we prefer to retain the pan-Arctic map in the original Figure 15. However, in response to the reviewer's comment, we have stripped it down to a single panel. We have also moved it to an "up front" position as the new Figure 1 in order to set the stage for our discussion of the role of landfast ice, especially with regard to the timing of break-up.*

Page 42:
"The sea ice indicators used here are based on local ice climatologies informed by community ice use" This is stated several times in the manuscript, but never discussed in detail. A reference to a previous publication is provided that perhaps used this community input. Two thoughts come to mind:
First, this needs more explanation here, given the number of modifications relative to the older work. IE were these modifications also "informed by community ice users?" In what way exactly?

*As explained in detail in Section 2 (Methods) of the paper, the modifications made relative to the prior study in the Alaskan Arctic were minor and meant to ensure broader applicability of thresholds, such as at locations further south where seasonal indicators need to be distinguished from fluctuations in ice edge position during the winter ice season.*

Second, is it true that this "community input" was solely from Alaskan local users? If yes, then is it good practice to develop sea ice diagnostics as done here and then apply them without modification to pan-Arctic coastal locations? This seems possibly ill-advised: any comments?

*The input to define key characteristics of the seasonal ice cycle was solely from Alaskan local users. However, the types of ice uses and ice hazards in this region, as reflected in the definition of key seasonal indicators, are of broader applicability to other locations in the Arctic. Specifically, the commonalities in coastal populations using coastal ice cover (both drifting and landfast) as a platform for a range of activities, and to whom sea ice poses a hazard for boat and marine vessel traffic, are such that they justify the approach taken in this study to extrapolate from the Alaskan Arctic (with a range of ice conditions representative of the broader Arctic) to the pan-Arctic scale. Revision of the text in the conclusions section in response to the reviewer's comments clarifies and provides further perspective on this point.*

Page 43: "from the author on request." Is data are available this sufficient for The Cryosphere?

*All or our computed dates are presented in the paper's figures (Figures 6-7. 9-11. 13). We could also include these vales in tables in the Supplementary Material section if the journal so desires. However, Supplementary Material is already table-heavy with its listings of slopes and significance levels, so we believe the additional listings would make the SM unwieldy with little value added.*

---

## Author Response (AR3)

**Responses to Reviewer 2 (3ʳᵈ round)** – Reviewer comment in regular font, *response in italics*

Line 302: There are perhaps ~ 6 uses of the term "fast ice" which should be changed to "landfast ice" for consistency and clarity.

*We have changed "fast ice" to "landfast ice" in the first five instances – lines 302, 331, 888, 891, and 1011.  However, in the 6ᵗʰ instance (line 1311), "fast ice" appears in the title of a paper in our reference list; we cannot change the title of a paper that is already published by other authors.*

Line 412: "dome" -> "done"

*Corrected.*